# Turbulent Characteristics of Saltation and Uncertainty of Saltation Model Parameters

Dongwei Liu[1], Masahide Ishizuka[2], Masao Mikami[3], Yaping Shao[4*]

[1]School of Ecology and Environment, Inner Mongolia University, China
liudw@imu.edu.cn
[2]Faculty of Engineering, Kagawa University, Japan
ishizuka@eng.kagawa-u.ac.jp
[3]Office of Climate and Environmental Research Promotion, Japan Meteorological Business Support Center, Japan
mikami@jmbsc.or.jp
[4]Institute for Geophysics and Meteorology, University of Cologne, Germany
yshao@uni-koeln.de

**Abstract:** It is widely recognized that saltation is a turbulent process, similar to other transport processes in the atmospheric boundary layer. Due to lack of high frequency observations, the statistic behavior of saltation is so far not well understood. In this study, we use data from the Japan-Australian Dust Experiment (JADE) to investigate the turbulent characteristics of saltation by analyzing the probability density function, energy spectrum and intermittency of saltation fluxes. Threshold friction velocity, $u_{*t}$, and saltation coefficient, $c_0$, are two important parameters in saltation models, often assumed to be deterministic. As saltation is turbulent in nature, we argue that it is more reasonable to consider them as parameters obeying certain probability distributions. The JADE saltation fluxes are used to estimate the $u_{*t}$ and $c_0$ probability distributions. The stochasticity of these parameters is attributed to the randomness in friction velocity and threshold friction velocity as well as soil particle size.

**Keywords:** wind erosion; turbulent saltation; saltation intermittency; saltation model; threshold friction velocity; saltation coefficient; maximum likelihood

**Highlight:** We use data from a field experiment to investigate saltation by analysing the probability density function, energy spectrum and intermittency of saltation fluxes. We also estimate two key wind-erosion model parameters and their probabilistic distributions. It continues the line of treating saltation as a turbulent process and represents a progress towards deriving more general wind erosion models.

## 1. Introduction

It is well-recognised that saltation, the hoping motion of sand grains near the earth's surface, is a turbulent process [Bagnold, 1941]. However, early studies focused mainly on its "mean" behaviour. Most well-known is, for example, the Owen [Owen, 1964] saltation model which predicts that the vertically integrated saltation flux is proportional to $u_*$ cubed, where $u_*$ is friction velocity, defined as $u_* = \sqrt{\tau / \rho}$ with $\tau$ being surface shear stress (N m$^{-2}$) and $\rho$ air density (kg m$^{-3}$). A dedicated investigation on turbulent saltation was conducted by Butterfield [1991], which revealed the significant variability of saltation fluxes concealed in conventional time-averaged data. Stout and Zobeck [1997] introduced the idea of saltation intermittency and pointed out that even when the averaged $u_*$ is below the threshold friction velocity, $u_{*t}$, saltation can still intermittently occur. The latter authors emphasized on saltation intermittency caused by fluctuations of turbulent wind, but stochasticity of $u_{*t}$ can also play a role. Turbulent saltation has attracted much attention in more recent years [e.g. McKenna Neuman et al. 2000; Davidson-

48 Arnott and Bauer, 2009; Sherman et al. 2017] and large-eddy simulation models have been
49 under development to model the process [e.g. Dupond et al. 2013]. However, due to a lack of
50 high-frequency field observations of saltation fluxes, the statistical behaviour of turbulent
51 saltation is, to date, not well understood.

52 A related problem is how saltation can be parameterized in wind erosion models. For example,
53 for dust modelling, it is important to quantify saltation, as saltation bombardment is a main
54 mechanism for dust emission. In wind erosion models, $u_{*t}$ is a key parameter which depends on
55 many factors including soil texture, moisture, salt concentration, crust and surface roughness.
56 In models, it is often expressed as

57

58 $$u_{*t}(d;\lambda,\theta,s_l,c_r,...) = u_{*t}(d)f_\lambda(\lambda)f_\theta(\theta)f_{sl}(s_l)f_{cr}(c_r)...\tag{1}$$

59

60 where $u_{*t}(d)$ is the minimal threshold friction velocity for grain size $d$ [Shao and Lu, 2000]; $\lambda$ is
61 roughness frontal-area index; $\theta$ is soil moisture; $s_l$ is soil salt content and $c_r$ is a descriptor of
62 surface crustiness; $f_\lambda$, $f_\theta$, $f_{sl}$ and $f_{cr}$ are the corresponding correction functions. The corrections
63 are determined semi-empirically, e.g., $f_\lambda$ using the Raupach et al. [1993] scheme and $f_\theta$ the Fécan
64 et al. [1999] scheme. The corrections $f_{sl}$ and $f_{cr}$ are so far not well known.

65

66 For homogeneous saltation, the saltation flux can be computed using the Kawamura [1964]
67 scheme, here multiplied by the fraction of erodible surface area $\sigma_f$,

68

69 $$Q(d) = \begin{cases} \sigma_f c_o \dfrac{\rho}{g}\ u_*^3\left(1-\dfrac{u_{*t}}{u_*}\right)\left(1+\dfrac{u_{*t}}{u_*}\right)^2 & u_* > u_{*t} \\ 0 & u_* \leq u_{*t} \end{cases}\tag{2}$$

70

71 where $d$ is particle diameter in sand particle size range and $g$ is acceleration due to gravity. The
72 saltation coefficient, $c_0$, is usually estimated empirically from field and/or wind-tunnel
73 experiments. It falls between 1.8 and 3.1 according to Kawamura [1964], and is commonly set
74 to 2.6 [White, 1979] in wind erosion models. The total (all particle sizes) saltation flux, $Q$, is a
75 particle-size weighted average of $Q(d)$

76

77 $$Q = \int_{d_1}^{d_2} Q(d)p_s(d)\delta d\tag{3}$$

78

79 where $d_1$ and $d_2$ define the upper and lower limits of saltation particle size, respectively, and
80 $p_s(d)$ is the soil particle size distribution. Observations show, however, $c_0$ varies considerably
81 from case to case (e.g. Gillette et al. 1997; Leys, 1998), and as the data presented later in this
82 paper show, for a given location, it may vary from day to day and even during a wind erosion
83 event.

84

85 While wind-erosion modules built in numerical weather and global climate models [e.g. Shao
86 et al. 2011; Kok et al. 2014; Klose et al. 2014] are in general more sophisticated than what is
87 described above and include a dust emission scheme, the estimate of $Q$ is essentially done using
88 Equations (1) to (3) or similar. Thus, the estimates of $u_{*t}$ and specification of $c_0$ are critical to
89 wind-erosion and dust modelling.


In most wind erosion models, both $u_{*t}$ and $c_0$ are treated as being deterministic. As saltation is
turbulent, it is more rational to treat $u_*$ and $c_o$ as parameters that satisfy certain probability
distributions. Saltation intermittency also implies that $u_{*t}$ and $c_0$ depend on the scale of
averaging. Shao and Mikami [2005] noticed that $u_{*t}$ for 10-minute averaged $Q$ and 1-minute
averaged $Q$ are quite different. Namikas et al. [2003] and Ellis et al. [2012] have also noticed
that averaging intervals of surface shear stress are important to quantifying sediment transport
because both shear stress and saltation flux are turbulent.
Between 23 Feb and 14 Mar 2006, Ishizuka et al. (2008; 2014) carried out the Japan-Australian
Dust Experiment (JADE) in Australia. In JADE, both $u_*$ and $Q$, together with a range of
atmospheric and soil surface quantities, were measured at relatively high sampling rates. The
loamy sand soil surface at the JADE site was very mobile and thus the JADE data are
representative to surfaces almost ideal for sand drifting. In this study, we analyse some aspects
of the turbulent behaviour of saltation using the JADE measurements of saltation fluxes. In light
of the analysis, we ask the question what the most likely values of $u_{*t}$ and $c_o$ are and how
representative they are. We also estimate the probability distribution of the two parameters.
**2. Data and Method for Parameter Estimation**
**2.1 JADE Data**
Ishizuka et al. carried out JADE between 23 Feb and 14 Mar 2006 on an Australian farm at
(33°50'42.4"S, 142°44'9.0"E). The size of field is about 1 km in the E–W direction and about
4 km in the N–S direction. A range of atmospheric variables, land surface properties, soil
particle-size distributions and size-resolved sand and dust fluxes were measured. During the
study period, 12 wind-erosion episodes were recorded. The dataset is particularly valuable in
that particle size resolved sand and dust fluxes [Shao et al. 2011] were measured. The details
of the experiments and datasets can be found in Ishizuka et al. [2008, 2014] and hence only a
brief summary is given here.
In JADE, three Sand Particle Counters (SPCs) [Yamada et al. 2002] were used to measure
saltation at the 0.05, 0.1 and 0.3 m levels with a sampling rate of 1 Hz. A SLD (Super
Luminescent Diode) light source is used to detect particles flying through the light beam. The
frequency of the input signal is 1-30 kHz, implying that particles moving with speed less than
30 m s$^{-1}$ can be detected. A SPC measures the saltation of particles in the range of 39 - 654 µm
in 32 bins with mean diameters of 39, 54, 69 µm etc. with irregular increment ranging between
15 and 23 µm. At each measurement height, the saltation flux density (M L$^{-2}$ T$^{-1}$), $q$, is obtained
as the sum of $q_j$ (saltation flux for size bin $j$) for the 32 size bins, i.e.
$$q = \sum_{j=1}^{32} q_j \tag{4}$$

The saltation flux, $Q$, is then estimated by integrating $q$ over height, namely,
$$Q = \int q\,dz \tag{5}$$

In computing $Q$, we assume $q = q_0 \exp(-az)$ with $q_0$ and $a$ being fitting parameters from the
measurements. Prior to the field experiment, the SPCs were calibrated in laboratory and during
JADE, they were checked in a mobile wind-tunnel at the site and compared with other saltation
samplers. But as $q$ was measured only at three heights, the vertical resolution of $q$ is relatively
poor and inaccuracies in the $Q$ estimates are unavoidable, which we are unable to fully quantify.
However, the profiles of $q$ are well behaved and thus the inaccuracies in the absolute values of
the $Q$ estimates are not expected to be so large as to affect the conclusions of this study.
$Q$ is computed using the SPC data at 1-second intervals. We denote its time series as $Q_{1sec}$.
From $Q_{1sec}$, the one-minute averages, $Q_{1min}$, and 30-minute averages of saltation fluxes, $Q_{30min}$,
are derived. All these quantities are also computed for individual particle size bins as
$$Q_j = \int q_j dz \qquad\qquad\qquad (5a)$$
Atmospheric variables, including wind speed, air temperature and humidity at various levels,
as well as radiation, precipitation, soil temperature and soil moisture were measured using an
automatic weather station (AWS). These quantities were sampled at 5-second intervals and their
averages over 1-minute intervals were recorded. Two anemometers were mounted at heights
0.53 m and 2.16 m on a mast for measuring wind speed. Also available are the Monin-Obukhov
length and sensible heat fluxes. From the wind measurements, surface roughness length $z_0$ and
friction velocity $u_*$ are derived, assuming a logarithmic profile (with stability correction) of the
mean wind. The roughness length for the experiment site is estimated to be 0.48 mm.
Friction velocity is computed with 1-minute averaged wind data, denoted as $u_{*1min}$, and 30-
minute averaged wind data, denoted as $u_{*30min}$. In atmospheric boundary-layer studies, there is
no standard for how long one should average wind to "correctly" estimate $u_*$, but it is common
to average over 10 to 30 minutes. But how long one averages depends on the purpose of the
averaging. If $u_*$ is used as a scaling velocity for the atmospheric boundary layer, e.g., as measure
of turbulence intensity, it is necessary to average over a sufficiently large time interval to obtain
a "constant" $u_*$. In this paper, $u_*$ is a surrogate of shear stress, the variation of which drives that
of saltation. Therefore, short averaging times are preferred, subject to that they are larger than
the response time of aeolian flux to shear stress. Anderson and Haff (1988) and Butterfield
(1991) suggested that this response time is of order of one second.
Observations of surface soil properties, including soil temperature and soil moisture, were made
at 1-minute intervals.  The surface at the JADE site was relatively uniform. A survey of ground
cover over an area of 900 x 900 m$^2$ at the site was made on 11 March 2006. The area was
divided into 9 tiles and surveyed along one transect of 300 m long in each tile. Photographs
were taken every 5 m by looking down vertically to a point on the ground. Surface cover was
estimated to be ~ 0.02 (see Appendix of Shao et al. 2011).
The wind erosion model, as detailed in Shao et al. (2011), is used for computing the saltation
fluxes using the JADE atmospheric and surface soil measurements as input. The saltation model
component is as described in Section 1, consisting of Equations (1) – (3). The fraction of
erodible surface area, $\sigma_f$, used in Equation (1), is one minus the fraction of surface cover. The
soil particle size distribution (psd), $p_s(d)$, required for Equation (3), is based on soil samples
collected at the JADE site and analyzed in laboratory. The analysis was done using a Microtrac
(Microtrac MT3300EX, Nikkiso Co. Ltd.), a particle size analyzer based on laser diffraction
light scattering technology. Water was used for sample dispersion. Depending on the methods
(pretreatment and ultrasonic vibration) used, the soil texture can be classified as sandy loam
(clay 0.3%, silt 25% and sand 74.7%) or loamy sand (clay 11%, silt 35% and sand 54%). The
sandy loam psd is used in this study, which has a mode at ~180 µm (see Shao et al. 2011, Fig.
5, Method A).

The default value of $c_0$ is set to 2.6, as widely cited in the literature [e.g. White, 1979] and the
default value of $u_{*t}$ is computed using Equation (1) with $u_{*t}(d)$ computed using the Shao and Lu
[2000] scheme, $f_\lambda$ using the Raupach et al. [1993] scheme, $f_\theta$ the Fécan et al. [1999] scheme,
and $f_{sl}$ and $f_{cr}$ set to one. The frontal area index $\lambda$ and soil moisture $\theta$ are both observed data
from JADE.
**2.2 Method for Parameter Estimation**
Different choices of $c_o$ and $u_{*t}$ would lead to different model-simulated saltation fluxes which
may or may not agree well with the measurements. By fitting the simulated saltation fluxes to
the measurements, we determine the optimal estimates of $c_0$ and $u_{*t}$ and the probability density
function (pdf) of these parameters. The method based on the Bayesian theory is used for the
purpose.
Suppose $\widetilde{X} = (\widetilde{x}_1, \widetilde{x}_2, ..., \widetilde{x}_n)$ is a measurement vector, with $\widetilde{x}_i$ being the measured value at time $t_i$,
and $A$ is a model with a forcing vector $F$ and model parameter vector $\beta$. Let the initial state of
the system be $i_0$, then the modelled value of the system, $X = (x_1, x_2, ..., x_n)$, can be expressed as
$$X(\beta) = A(i_0, F; \beta) \qquad\qquad (6)$$
The error vector is given by $E(\beta) = \widetilde{X} - X$, here, fully attributed to $\beta$. Given $\widetilde{x}$, the posterior
parameter pdf, $p(\beta|\widetilde{X})$, can be estimated from the Bayes theorem:
$$p(\beta|\widetilde{X}) \propto p(\beta)p(\widetilde{X}|\beta) \qquad\qquad (7)$$
where $p(\beta)$ is the prior parameter pdf and $p(\widetilde{X}|\beta)$ the likelihood. If $p(\beta)$ is given, then the
problem of finding $p(\beta|\widetilde{X})$ reduces to finding the maximum likelihood. Assuming the error
residuals are independent and Gaussian distributed with constant variance, $\sigma^2$, the likelihood
can be written as
$$p(\widetilde{X}|\beta) = \prod_{i=1}^{n} \frac{1}{\sqrt{2\pi}\sigma} \exp\left(-\frac{(x_i - \widetilde{x}_i)^2}{2\sigma^2}\right) \qquad\qquad (8)$$
In this case, maximizing the likelihood is equivalent to minimizing the error, i.e.,
$$R^2(\beta) = \min \sum_i (x_i - \widetilde{x}_i)^2 \qquad\qquad (9)$$
The solution of Equation (9) gives an optimal (i.e. with maximum likelihood) estimate of mean
$\beta$. This is the popular least-squares method. A disadvantage of the method is that it assumes a
Gaussian posterior parameter pdf and the computing the $\beta$ variance requires the pre-knowledge
of the accuracy of the data.
As an alternative, the approximate Bayesian computation (ABC) method has been proposed
[e.g. Vrugt and Sadegh, 2013]. It is argued that a parameter value $\beta^*$ should be a sample from
$p(\beta|\widetilde{X})$ as long as the distance between the observed and simulated data is less than a small
positive value

$$\rho(\beta^*) = |X(\beta^*) - \widetilde{X}| \le \varepsilon \qquad\qquad (10)$$

This procedure provides explicitly an estimate of parameter pdf for given dataset. The ABC
method is numerically simple: from a prior pdf (e.g. uniform) of $\beta$ a $\beta^*$ is stochastically
generated and the model is run. If Equation (10) is satisfied, then $\beta^*$ is accepted or otherwise
rejected. This procedure is repeated and the a-priori pdf of $\beta$ is mapped to a posterior pdf of $\beta$.
The ABC method has the disadvantage though that it is numerically inefficient. More efficient
techniques based on the same principle exist, e.g., Markov Chain Monte Carlo Simulation
[Sadegh and Vrugt, 2014]. In this study, we apply the Differential Evolution Adaptive
Metropolis (DREAM) algorithm proposed by Vrugt et al. (2011) for estimation of hydrologic
model parameters. The algorithm integrates Differential Evolution [Storn and Price, 1997] and
self-adaptive randomized subspace sampling to accelerate a Markov Chain Monte Carlo
simulation. A full description of the DREAM algorithm is beyond the scope of our study.
Interested readers should refer to the above cited references for details.

**3. Statistical Features of Saltation**

**3.1 Time Series**

To provide an overview of the dataset used in this study. Fig. 1a shows the time series of $Q_{1min}$
and $u_{*1min}$, and Fig. 2 $Q_{30min}$ and $u_{*30min}$. During the 20-day period, aeolian sand drift occurred
almost every day at the site according to the field logging book, but only 12 events were
recorded using the SPCs. Saltation fluxes were not measured on Day 55, 58, 59, 64 and then
Day 66 to 70, due to either instrument maintenance or use of the SPCs for other purposes (e.g.
wind-tunnel experiments). The figures show that both $Q$ and $u_*$ fluctuate significantly and
saltation is turbulent. Fig. 1b shows an enlarged plot of the $Q_{1min}$ and $u_{*1min}$ time series for Day
61 and 62. At the JADE site, $u_{*t}$ was about 0.2 m s$^{-1}$. On Day 61, $u_*$ was mostly larger than this
value and saltation was almost continuous, while on Day 62, $u_*$ was close to this value and
weak saltation occurred frequently also when $u_*$ was below 0.2 m s$^{-1}$. Fig. 2b is as Fig.1b, but
for $Q_{30min}$ and $u_{*30min}$. A comparison of Fig. 1b and Fig. 2b reveals that the amplitude of the
$Q_{1min}$ fluctuations is several times of that of the $Q_{30min}$ fluctuations. A strong correlation between
the time series of $Q_{30min}$ and $u_{*30min}$ can be directly seen in Fig. 2b.

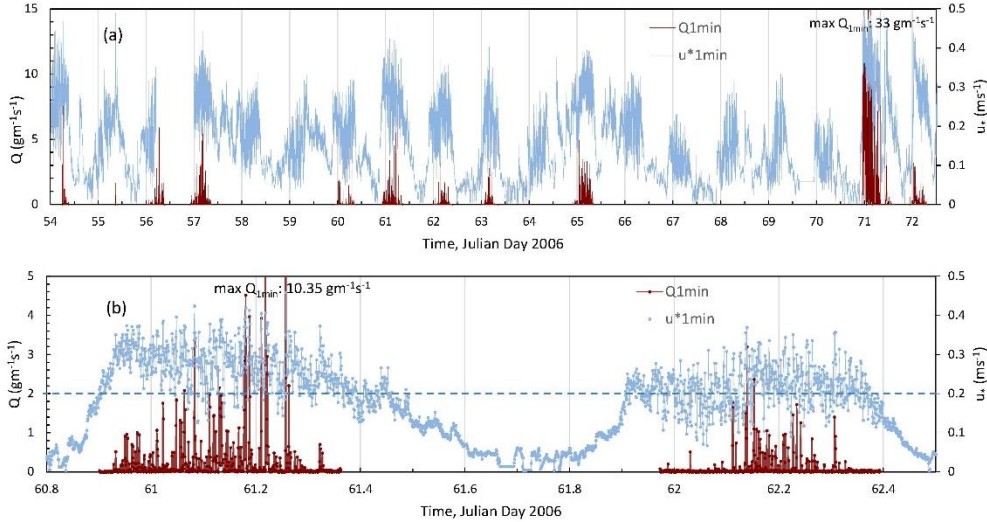

Figure 1: (a) Observed time series of 1-min averaged saltation flux, $Q_{1min}$ (g m$^{-1}$ s$^{-1}$), and friction
velocity, $u_{*1min}$ (m s$^{-1}$), for the JADE study period; (b) an enlarged plot of (a) for the erosion
events on Day 61 and 62. Note that the axes in (b) have different scales than in (a).

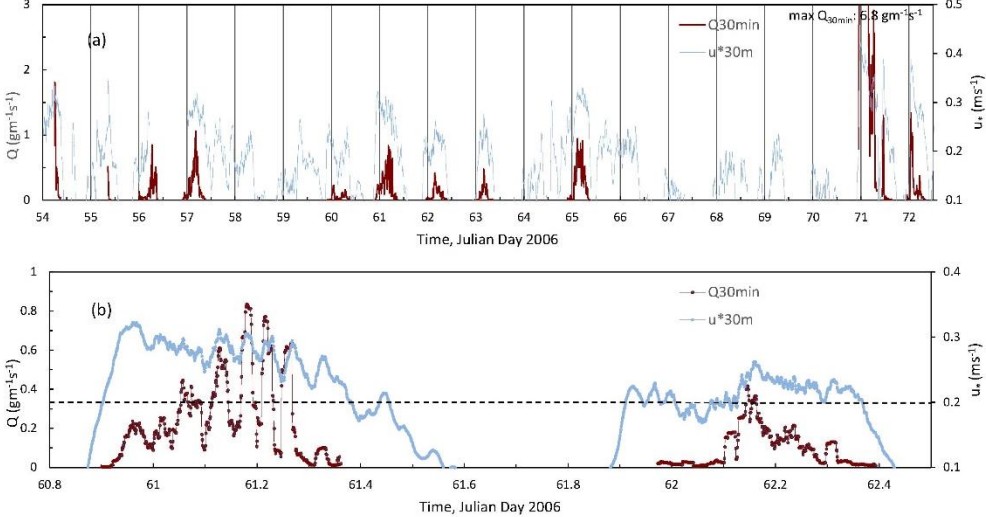

Figure 2: As Fig. 1, but for running means over 30-min intervals.
In Fig. 3a, b and c, $Q$ is plotted against $u_*{}^3$. Several interesting features can be identified. For
the majority of the points, the $Q \sim u_*{}^3$ relationship appears to hold, but this relationship can vary
significantly even for the same data set from event to event. For example, large differences exist
between days 70 and 71 (denoted D70-71, an event of intensive wind erosion) and Day 72 (a
day of weak wind erosion), as seen in both Fig. 3a and Fig. 3b. There may be many likely
reasons for the differences the $Q \sim u_*$ relationship but the most conspicuous are differences in
atmospheric turbulence (e.g., gustiness) and time-varying surface conditions (e.g. particle
sorting and aerodynamic roughness). Fig. 3d shows the time series of $(u_{*1min}-u_{*30min})$, a measure
of turbulent fluctuations. It is seen that saltation is associated with not only high surface shear
stress but also high shear stress fluctuations. The large difference in the $Q \sim u_*$ relationship
between D70-71 and D72 (Fig. 3b) is probably attributed to the strong differences in turbulent
fluctuations (Fig. 3d): D70-71 was a hot gusty day with top (2 cm) soil temperature reaching
53$^{o}$C, while D72 was cooler and less gusty with soil temperature 5$^{o}$C lower. Also hysteresis is
observed in the $Q \sim u_*$ relationship, as shown in Fig. 3c, using D71 and D72 as example. Fig. 3d
shows that for all three events selected (D70-71, D71 and D72), saltation has a relatively short
(0.5 to 2 hours) strengthening phase, followed by a longer weakening phase. During an erosion
event, for the same $u_*$, saltation is stronger in the strengthening than in the weakening phase.
An examination of Fig. 3d suggests that the hysteresis cannot be simply attributed to the
intensity of turbulence. We speculate that it is probably more related to flow-saltation feedbacks
(e.g. stronger splash entrainment in the strengthening phase) and the modification of surface
aerodynamic conditions (e.g. particle sorting and reduced surface roughness Reynolds number).

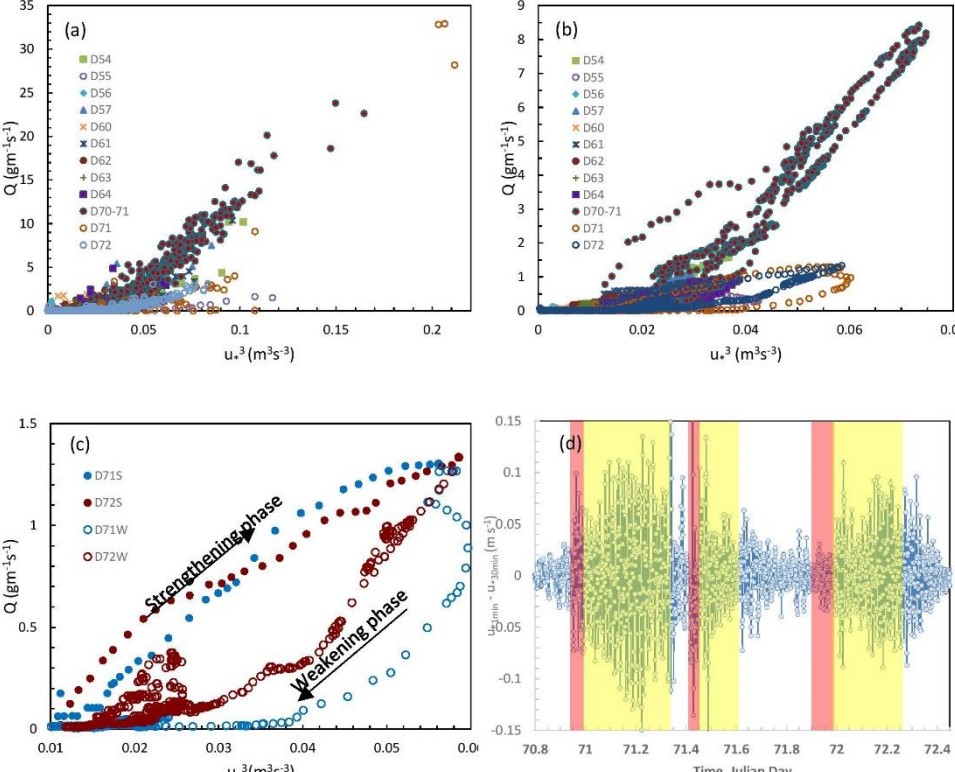

Figure 3: (a) Saltation flux, $Q$ (g m$^{-1}$ s$^{-1}$), plotted against friction velocity, $u_*^3$ (m$^3$ s$^{-3}$), for 1-
minute averages; (b) As (a), but for 30-minute averages; (c) As (b), but enlarged to illustrated
saltation hysteresis on D71 and 72; D71S/72S denote the strengthening and D71W/72W the
weakening phase of the D71/72 event; (d) Time series of $u_*$ derivations, given by ($u_{*1min} - u_{*30min}$),
for D70-71, D71 and D72. The strengthening phase is marked red and the weakening phase
yellow.
**3.2 Probability Density Function of Saltation Fluxes**
How well the saltation model performs, whether $u_{*t}$ and $c_o$ are universal and how they are
probabilistically distributed must depend on the turbulent properties of saltation. As the JADE
saltation fluxes are sampled at 1 Hz, we can use these data to examine (to some degree) the
statistical behavior of saltation. In Fig. 4, the pdfs of the saltation fluxes for different particle
size groups are plotted, computed using $Q_{1sec}$ and $Q_{1min}$. It is seen that the pdfs generally behaves
as
$$p(Q) \propto Q^{-\alpha} \qquad\qquad (11)$$

In case of $Q_{1sec}$, there seems to be a distinct change in $\alpha$ at a critical value of $Q_c \sim 3$ g m$^{-1}$ s$^{-1}$,
with $\alpha \sim 1$ for $Q < Q_c$ and $\alpha \sim 4$ for $Q > Q_c$. The pdfs derived from $Q_{1min}$ appear to follow the
basic functional form of Equation (11). Again, $\alpha$ is about 1 and tends to be larger for large $Q$
values. Fig. 4 shows that the pdfs of $Q$ depend significantly on the interval of time averaging,
i.e., after averaging, smaller saltation fluxes become more frequent. This is because the time
series of $Q_{1sec}$ is more intermittent (see also Fig. 6).

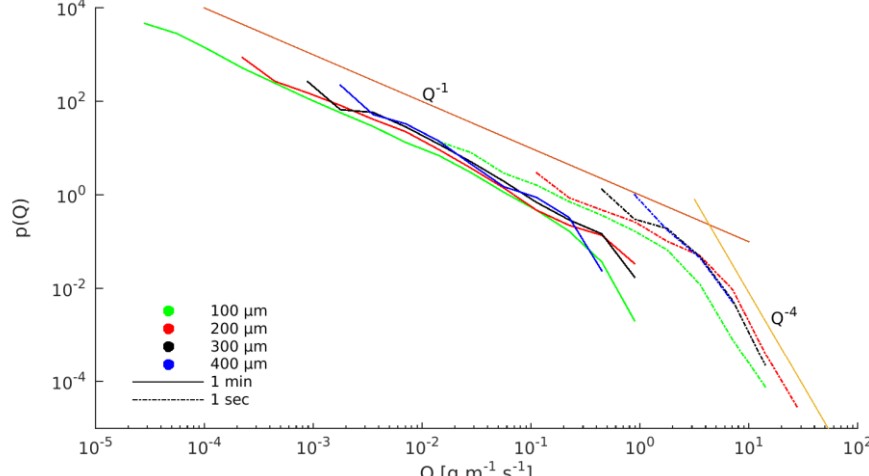

Figure 4: Probability density functions of $Q_{1sec}$ (solid lines) and of $Q_{1min}$ (dashed lines) for four
different particle sizes. Two additional lines $p(Q) \sim Q^{-1}$ and $Q^{-4}$ are drawn as reference.
The pdfs of $Q_{1sec}$ and $Q_{1min}$ integrated over all particles are shown in Figure 5b. Again, the pdfs
show the general behavior of $p(Q) \sim Q^{-1}$. In theory, $p(Q)$ can be derived from the pdf of $u_*$,
$p(u_*)$. From Equation (2), we have

$$\frac{dQ}{du_*} = c_0 \frac{\rho}{g}\left(3u_*^2 + 2u_*u_{*t} - u_{*t}^2\right) \quad \text{for} \quad u_* > u_{*t} \tag{12}$$
This can be used to obtain

$$p(Q) = \begin{cases} p(u_*)\dfrac{du_*}{dQ} & \text{for} \quad u_* > u_{*t} \\ 0 & \text{for} \quad u_* \leq u_{*t} \end{cases} \tag{13}$$
Fig. 5a shows the $p(u_*)$ estimated from $u_{*1min}$ together with the fitted Weibull distribution. For
the fitting, emphasis is made to ensure that $p(u_*)$ for $u_* > 0.2$ ms$^{-1}$ is best approximated. Fig. 5b
shows the $p(Q)$ estimated from $Q_{1min}$. We computed $p(Q)$ using Equation (13) with the fitted
$p(u_*)$, assuming $u_{*t} = 0.2$ ms$^{-1}$ and $c_0 = 2.6$. It is seen that the observed and modelled $p(Q)$ have
qualitative similarities (namely $p(Q)$ decreases with increasing $Q$) but using Equations (12) and
(13) we cannot well reproduce the observed $p(Q)$. For example, the model fails to predict the
lowly frequent strong saltation fluxes and fails to predict the highly frequent weak saltation
occurring when $u_*$ is below the specified threshold. Tests using several smaller $u_{*t}$ values (0,
0.05 and 0.1). With smaller $u_{*t}$ values, the highly frequent weak saltation fluxes are better
reproduced, but fra from satisfactory.


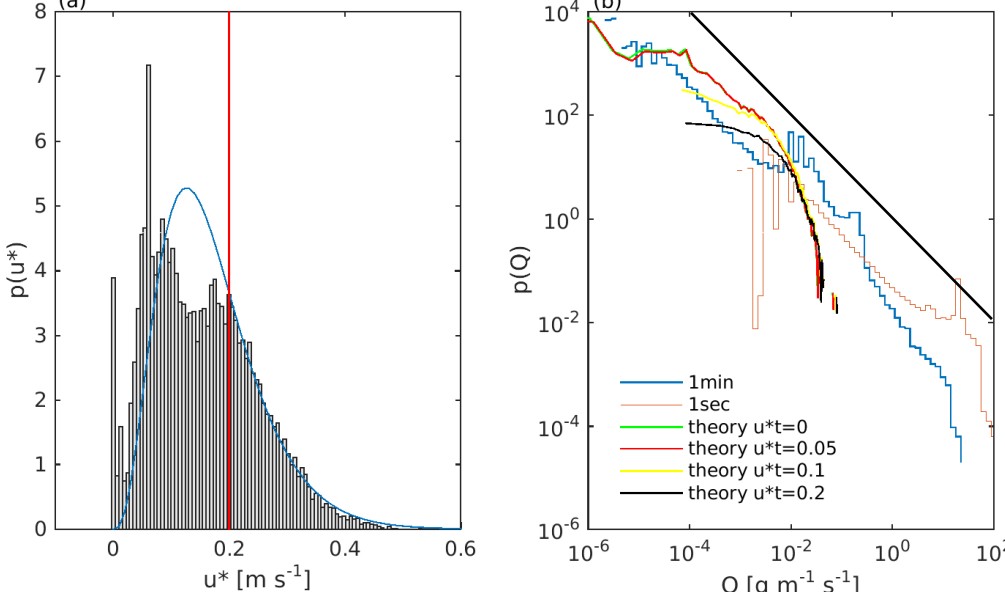

Figure 5: (a) Probability density functions of friction velocity, $p(u_*)$, plotted against $u_*$ (bars).
To compute $p(u_*)$, $u_{*1min}$ is used; a Weibull distribution (blue line) is fitted to $p(u_*)$; the red line
marks the assumed threshold friction velocity. (b) Probability density function of $Q$, $p(Q)$,
estimated using $Q_{1min}$ (blue) and $Q_{1sec}$ (dark red) and using Equation (13) assuming several u*t
values ($u_{*t} = 0.0$ m s$^{-1}$, green; 0.05 m s$^{-1}$, red; 0.1 m s$^{-1}$, yellow; 0.2 m s$^{-1}$, black). The $p(Q) \sim Q^{-1}$
$^1$ line is also drawn for comparison.
**3.3 Saltation Intermittency**
Following Stout and Zobeck [1997], the intermittency of saltation, $\gamma$, is defined as the fraction
of time during which saltation occurs at a given point in a given time period. It should be pointed
out that as saltation is a turbulent process, saltation intermittency describes only the behaviour
of the process at $u_* \sim u_{*t}$, i.e., saltation intermittency is merely a special, although important,
case of turbulent saltation. Several formulations of $\gamma$ are possible. Stout and Zobeck [1997]
assumed that saltation occurs only in time windows when $u_*$ exceeds $u_{*t}$. Therefore, if $p(u_*)$ is
known, then $\gamma$ for a given $u_{*t}$ can be estimated as
$\gamma_a(u_{*t}) = 1 - \int_0^{u_{*t}} p(u_*)du_*$  (14a)
Stout and Zobeck [1997] used the counts per second of sand impacts on a piezoelectric crystal
saltation sensor as a measure of saltation activity and found that $\gamma_a$ rarely exceeded 0.5.
In Equation (14a) $u_{*t}$ is fixed and thus saltation intermittency is attributed entirely to the
fluctuations of $u_*$. In reality, $u_{*t}$ also fluctuates and satisfies certain pdfs (Raffaele et al., 2016).
In analogy to Equation (14a), $\gamma$ for a given $u_*$ can be estimated as
$\gamma_b(u_*) = 1 - \int_{u_*}^{\infty} p(u_{*t})du_{*t}$  (14b)
More generally, we can define saltation intermittency as

$\gamma_c = \int_0^\infty \left[1 - \int_0^{u_{*t}} p(u_*) du_*\right] p(u_{*t}) du_{*t} = \int_0^\infty \gamma_a(u_{*t}) p(u_{*t}) du_{*t}$      (14c)

or

$\gamma_c = \int_0^\infty \left[1 - \int_{u_*}^\infty p(u_{*t}) du_{*t}\right] p(u_*) du_* = \int_0^\infty \gamma_b(u_*) p(u_*) du_*$      (14d)

Equations (14c) and (14d) reduce to Equation (14a) in case of $p(u_{*t}) = \delta(u_{*t})$ and to Equation
(14b) in case of $p(u_*) = \delta(u_*)$, respectively.

The computation of saltation intermittency function $\gamma_a(u_{*t})$ is done by integrating $p(u_*)$ (Fig. 5a)
to fixed value of $u_{*t}$. In Fig. 6a, $\gamma_a$ as function of $u_{*t}$ is plotted. The behaviour of $\gamma_a(u_{*t})$ is as
expected: it is one at $u_{*t} = 0$ and decreases to zero at about $u_{*t} = 0.5$ ms$^{-1}$ as in the case of JADE,
$u_*$ rarely exceeded this value. For $u_{*t} = 0.2$ ms$^{-1}$, $\gamma_a$ is 0.35, comparable with the result of Stout
and Zobeck (1997) who reported an intermittency of 0.4. As $p(u_{*t})$ is not known, Equation (14b)
cannot be used directly, but we can compute $\gamma_b(u_*)$ using the JADE data. First, it is computed
using $Q_{1min}$. This is done by selecting a fixed $u_*$ say $u_{*c}$, and counting the time fraction, $T_{u*}$,
which satisfies $|u_* - u_{*c}| < \delta$ (used is $\delta = 0.05$ ms$^{-1}$) and the time fraction, $T_{Q1min}$, which
satisfies $|u_* - u_{*c}| < \delta$ and $Q_{1min} > 0$. By definition, saltation intermittency is $T_{Q1min}/T_{u*}$ as
plotted in Fig. 6a. It is seen that for $Q_{1min}$, $\gamma_b(u_*)$ increases from about 0.6 at $u_* \sim 0.1$ ms$^{-1}$ to
about one at $u_* = 0.3$ ms$^{-1}$. This shows that in JADE a considerable fraction of the saltation
fluxes was recorded at u* below the perceived threshold friction velocity (about 0.2 ms$^{-1}$),
saltation is more intermit under weak wind conditions and becomes non-intermittent for $u_* >$
0.3 ms$^{-1}$. The increase of $\gamma_b(u_*)$ with decreasing $u_*$ for $u_* < 0.1$ ms$^{-1}$ is however unexpected. The
expected $\gamma_b(u_*)$ for small $u_*$ is as depicted using the dashed line. A likely reason for the
unexpected behaviour of $\gamma_b(u_*)$ is that during a wind erosion event, grains in saltation may
continue to hop even when $u_*$ is temporarily reduced to small values. The uncertainty in the
data also needs to be considered, as the sample size for determining the ratio $T_{Q1min}/T_{u*}$ becomes
smaller. More complete datasets are required to answer these questions. Finally, $\gamma_c$ is computed
by using Equation (14d) and is found to be around 0.73. For the one-second case, we cannot
plot $\gamma_b$ as a function of $u_*$, because $u_*$ is not available at such high frequency. We computed $\gamma_c$
for individual particle size groups (Fig. 6b) using $Q_{1sec}$, $Q_{1min}$ and $Q_{30min}$, which is the time
fraction of saltation for a given particle size, $d$, during the saltation event. It is found that $\gamma_c(d)$
decreases with $d$, i.e., the saltation of larger particles is more intermittent. Also, $\gamma_c(d)$ increases
with increased averaging time intervals, implying that the small scales features of turbulence
play an important role in intermittent saltation.

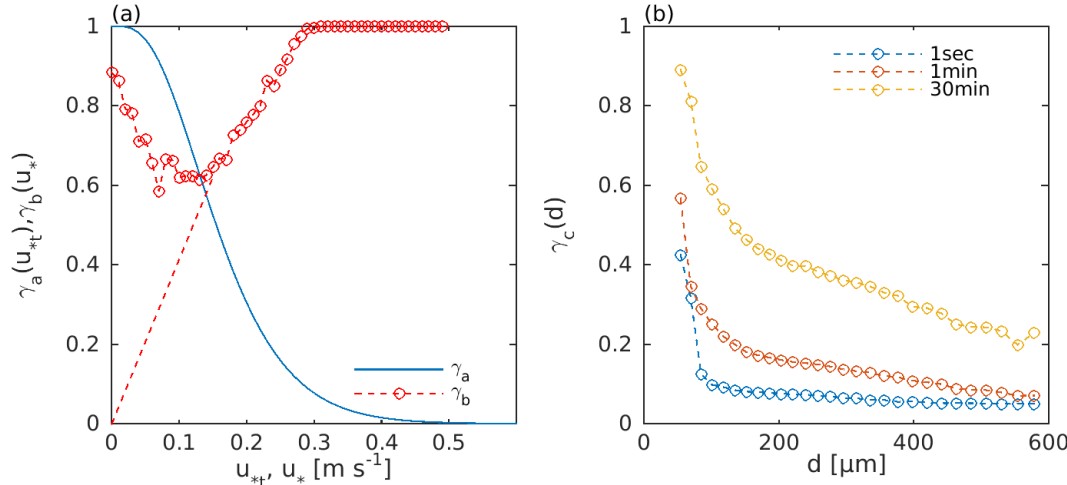

Figure 6: (a) Saltation intermittency function $\gamma_a(u_{*t})$, and $\gamma_b(u_*)$. See text for more details. (b) $\gamma_c$ as a function of particle size for $Q_{1sec}$, $Q_{1min}$ and $Q_{30min}$.

### 3.4 Spectrum of Saltation Fluxes

Spectral analysis is a widely used for characterising the variations of a stochastic process on different scales. Using the JADE data, we computed the power spectrum of saltation fluxes, $P_Q(f)$ at frequency $f$, and of friction velocity, $P_{u*}(f)$, using a non-uniform discrete Fourier transform. For comparison, the power spectra are normalized with the respective variances of the signal. In atmospheric boundary-layer studies, the spectra of various turbulence quantities have been thoroughly investigated (Stull, 1988). Examples for spectra of Reynolds shear stress can be found in McNaughton and Laubach (2000). Fig. 7 shows $P_Q(f)$ and $P_{u*}(f)$ (Fig. 7a) as well their co-spectrum (Fig. 7b). $P_Q(f)$ is computed using both $Q_{1sec}$ and $Q_{1min}$, and $P_{u*}(f)$ with $u_{*1min}$. It is seen that the power spectra of $Q$ and $u_*$ have qualitatively very similar behaviour. Both have a maximum at about $10^{-5}$ Hz, a minimum at about $10^{-4}$ Hz and another peak at about $2 \times 10^{-3}$ Hz. The maximum at $10^{-5}$ Hz is related to the diurnal patterns and changing synoptic events, which drive the wind erosion episodes, the minimum at $10^{-4}$ Hz is due to the lack of turbulent winds at the time scale of several hours, while the peak at $2 \times 10^{-3}$ Hz is caused by the minute-scale gusty winds/large eddies in turbulent flows. Also the $Q$-$u_*$ co-spectrum shows that $Q$ and $u_*$ are most strongly correlated on diurnal/synoptic and gust/large-eddy time scales. $P_Q(f)$ computed using $Q_{1sec}$ reveals again the peaks at $10^{-5}$ Hz and at $2 \times 10^{-3}$ Hz. The power of the $Q$ spectrum then decreases with frequency. As the sampling rate of saltation flux is limited to one second in this study, the features of $P_Q(f)$ at frequencies larger than 0.5 Hz are not resolved.

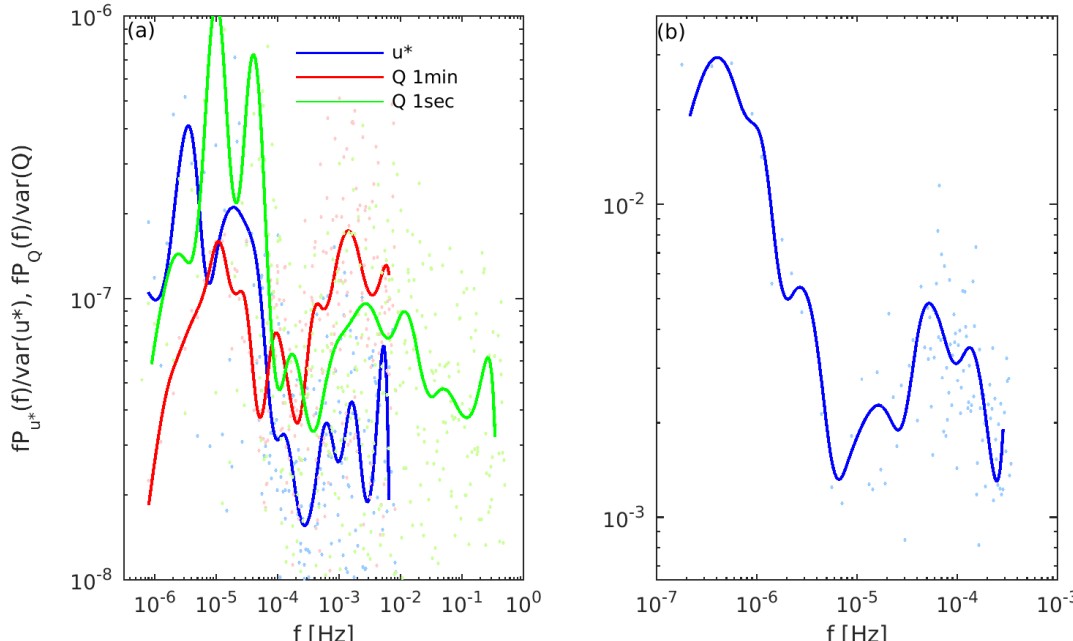

Figure 7: (a) Normalized power spectrum of $u_*$(blue) computed with $u_{*1min}$, together with the
normalized power spectrum of saltation flux computed with $Q_{1min}$ (red) and $Q_{1sec}$ (green). (b)
Normalized $Q$-$u_*$ co-spectrum, computed using with $Q_{1min}$ and $u_{*1min}$. In both (a) and (b), dots
are unsmoothed spectra, and curves are smoothed spectra.
**4. Estimates of Saltation Model Parameters**
Given the turbulent nature of saltation, it is rational to treat $u_{*t}$ and $c_0$ in the saltation model as
parameters obeying certain probability distributions. To examine the behavior of these
parameters, we introduce two coefficients $r_{c0}$ and $r_{u*t,}$ and multiply them respectively by the
"theoretical" values of $c_0$ and $u_{*t}$ in Equation (2), i.e.
$$u_{*t} = r_{u*t}u_{*t,theory}$$
$$c_0 = r_{c0}c_{0,theory}$$

As introduced in Section 1, we assumed $c_{0,\ theory} = 2.6$ and computed $u_{*t,\ theory}$ using Equation (1)
with observed soil moisture and fraction of cover. The two coefficients $r_{c0}$ and $r_{u*t}$ are varied to
generate a model estimate of $Q$ using Equations (2) and (3) with observed $u_*$. The probability
distributions of $r_{c0}$ and $r_{u*t}$ are estimated using the following techniques. Let us denote the time
series of the modelled saltation flux as $Q_{M,i,}$ ($i=1,N$) and of the corresponding measurement $Q_{D,i}$.
The absolute error, $\delta Q_A$, and Nash coefficient, $I_{Nash}$, are used as measures for the goodness of
the agreement between the model and the measurement. They are defined as,
$$\delta Q_A = \frac{1}{N}\sum|a_i|$$
$$I_{Nash} = (1 - \sum a_i^2 / \sum b_i^2)$$

with

$$a_i = Q_{M,i} - Q_{D,i}$$

$$b_i = Q_{M,i} - \frac{1}{N}\sum Q_{M,i}$$

$$c_i = \begin{cases} a_i / Q_{M,i} & Q_{M,i} \neq 0 \\ 0 & \text{else} \end{cases}$$

The prior pdfs of $r_{c0}$ and $r_{u*t}$ are assumed to be uniform. In the numerical experiment, we
randomly generate $r_{c0}$ and $r_{u*t}$ and seek their values, such that $\delta Q_A \leq \varepsilon$ and $I_{Nash} > \eta$. These
experiments are repeated for $Q_{1min}$ and $Q_{30min}$. The plots of $\delta Q_A$ and $I_{Nash}$ as functions of $r_{c0}$ and
$r_{u*t}$ show that for certain values of $r_{c0}$ and $r_{u*t}$, the above conditions are satisfied. Fig. 8 shows
that for $Q_{1min}$, the best simulation is achieved with $r_{c0} = 1.23$ and $r_{u*t} = 1.05$, while for the $Q_{30min}$,
with $r_{c0} = 0.94$ and $r_{u*t} = 0.91$. This suggest that the "optimal" estimates of $u_{*t}$ and $c_0$ are close
to the corresponding theoretic values, but are dependent on the time averaging intervals, with
both $u_{*t}$ and $c_0$ being larger for shorter averaging intervals.

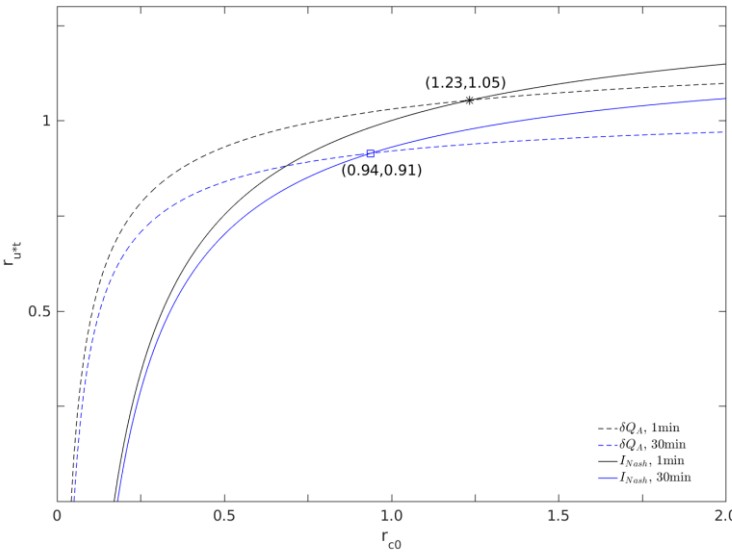

Figure 8: $\delta Q_A$ and $I_{Nash}$ are both functions of $r_{c0}$ and $r_{u*t}$. Along the dashed curves, the
condition $\delta Q_A = \min$ is satisfied and along the solid curves the condition $I_{Nash} = \max$ is
satisfied. The curves are estimated with both $Q_{1min}$ and $Q_{30min}$.
The parameter pdfs $p(r_{u*t})$ and $p(r_{c0})$ are estimated with the DREAM algorithm, again using the
absolute error and the Nash coefficient as goodness of agreement between the model simulated
and measured saltation fluxes. The results are shown in Fig. 9. All pdfs are fitted to a $\Gamma$-
distribution. As seen in Fig. 9a and 9c, the most frequent $r_{u*t}$ values are respectively 1.12 and
1.04 for $Q_{1min}$ and $Q_{30min}$, close to the estimates of 1.05 and 0.91 found in Fig. 8. For $Q_{1min}$, $r_{u*t}$
is ~$1.12 \pm 0.2$ and for $Q_{30min}$ ~$1.04 \pm 0.3$. This implies that sometimes saltation occurs when $u_*$
is below the theoretical $u_{*t}$ value and sometimes saltation does not occur even when $u_*$ is above
it, as already seen in Fig. 6a. In the case of $p(r_{c0})$ (Fig. 9c and 9d), the most frequent values of
$r_{c0}$ for $Q_{1min}$ and $Q_{30min}$ are, respectively, 1.04 and 0.92, close to the optimal estimates of 1.23
and 0.94 shown in Fig. 8. But $r_{c0}$ varies over a wide range, for instance, for $Q_{30min}$ between 0.5
and 5, i.e., $c_0$ is a rather stochastic parameter.

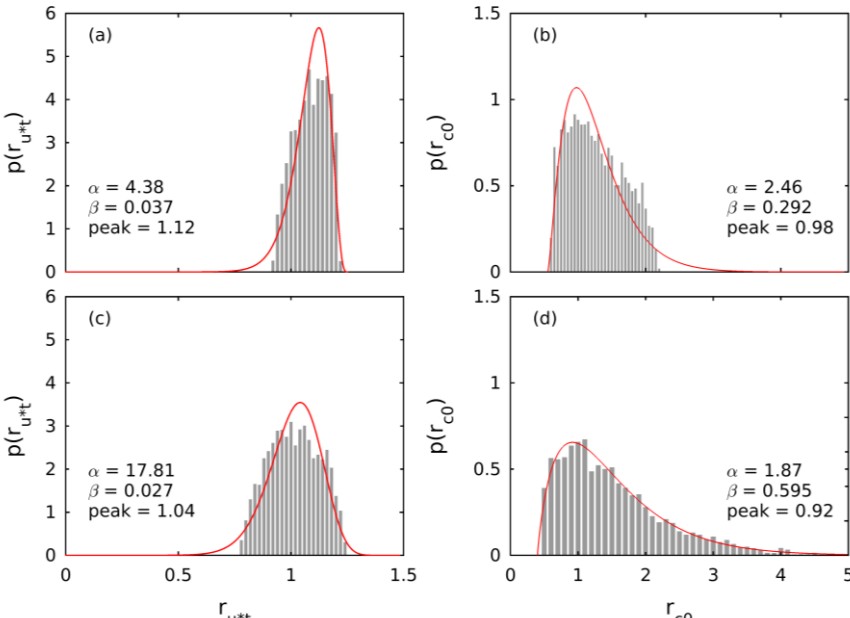

Figure 9: (a) Parameter pdf $p(r_{u*t})$ for 1-min averaged saltation fluxes; (b) as (a), but for $p(r_{c0})$; (c) and (d), as (a) and (b), but for 30-min averaged saltation fluxes.

In nature, many factors influence sediment transport, but the stochasticity of the parameters is determined primarily by the turbulent fluctuations of friction velocity (or surface shear stress), the randomness of threshold friction velocity, and soil particle size distribution (representing particle response to forcing). Studies have shown, for instance, that small changes in soil moisture can have large influences on saltation [Ishizuka et al. 2008] and soil moisture in the very top soil layer can vary significantly over relatively short time periods. Over the period of 18 days during JADE soil moisture in the top 0.05 m layer varied between 0.02 and 0.04 $m^3 m^{-3}$ (4 and 8% in relative soil moisture, assuming a saturation soil moisture of 0.5 $m^3 m^{-3}$). In this study, the influence of soil moisture on saltation is accounted for via Equation (1) using the soil moisture measurements in the top 0.05m layer (see also Fig. 4a in Shao et al. 2011). While measured soil moisture is used in the wind erosion model, the randomness associated with its spatial-temporal variations is not, which is most likely reflected in the stochasticity of $u_{*t}$.

The stochasticity of $c_0$ arises because saltation fluctuates, depending on turbulence and particle size. To demonstrate this, we divided the time series of the saltation fluxes into two subsets, one with $Q_{D,i} \leq 3$ g $m^{-1}$ $s^{-1}$ representing weak saltation and one with $Q_{D,i} > 3$ g $m^{-1}$ $s^{-1}$ representing significant saltation. This separation is arbitrary but sufficient for making the point that $c_0$ depends on $u_*$, also a measure of turbulence intensity. The parameter pdfs, $p(r_{u*t})$ and $p(r_{c0})$, for the subset $Q_{D,i} \leq 3$ g $m^{-1}$ $s^{-1}$ is shown in Fig. 10. For $Q_{1min}$ and $Q_{30min}$, the most frequent $r_{u*t}$ values are now respectively 0.99 and 0.85, somewhat smaller than the estimated values for the full set (Fig. 9). In comparison, the most frequent $r_{c0}$ values are now respectively 0.30 and 0.29, three to four times smaller than for the case when the full set is considered (Fig. 9). This suggests that $c_0$ has a clear dependency on $u_*$ and is smaller for smaller $u_*$. This is because saltation is more intermittent in the case of smaller $u_*$ (i.e. smaller excess shear stress) and thus, $c_0$, a descriptor of the relation between time-averaged saltation flux and friction velocity, is smaller for more intermittent saltation.

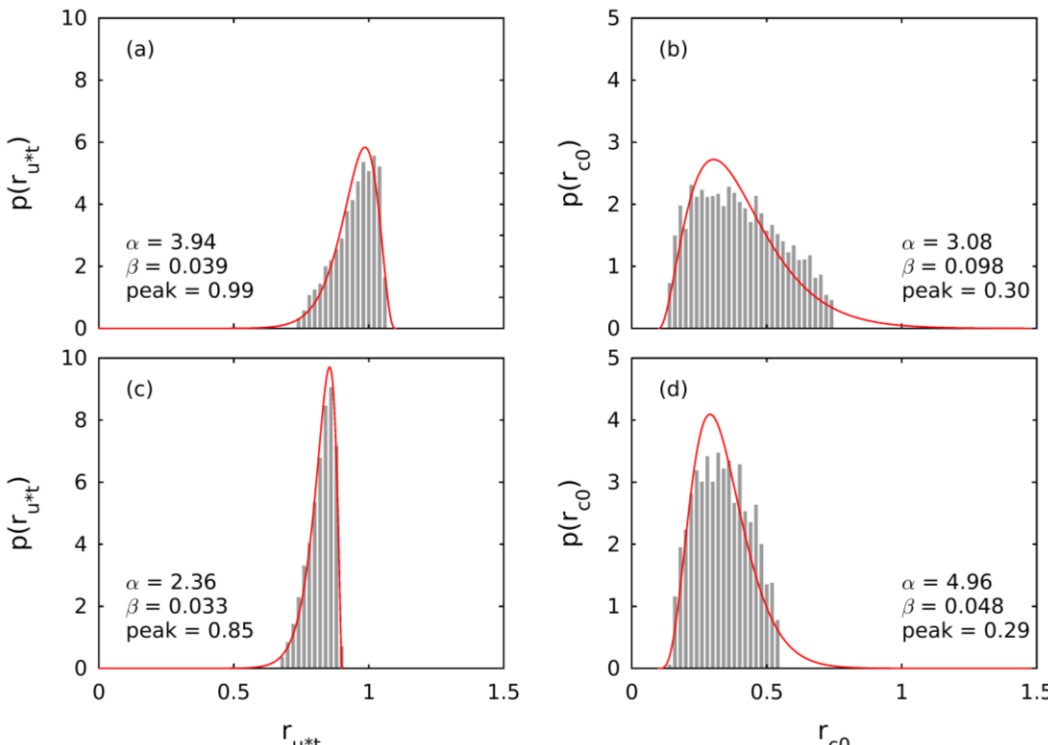

Figure 10: As Fig. 9, but estimated using the time series of saltation fluxes which satisfy $Q_{D,i} \leq$
$3$ g m$^{-1}$ s$^{-1}$.
We fit the pdfs, $p(r_{u*t})$ and $p(r_{c0})$, for individual particle size bins and found that the most
frequent $r_{u*t}$ values do not differ substantially among the particle sizes, but $r_{c0}$ depends
systematically on particle size. For example, the most frequent $r_{c0}$ values for 101, 151, 203, 315
and 398 µm are, respectively, 0.5, 1.3, 1.7, 3.1 and 4.0. These values are obtained by first
estimating $p(r_{c0})$ for the individual particle size bins with the measured saltation flux for the
corresponding bins and then normalizing $p(r_{c0})$ with the mass fraction of the size bins of the
parent soil. A least squares curve fitting shows that the most frequent $r_{c0}$ value depends almost
perfectly ($R^2 = 0.996$) linearly on particle size:
$$r_{c_0} = 0.012d - 0.59 \qquad\qquad (15)$$
for the particle size range (100 to 400 µm) we tested, with $d$ being particle size in µm.
We have shown that both $u_{*t}$ and $c_0$ satisfy certain pdfs that depend on the properties of the
surface, atmospheric turbulence and soil particle size. Fig. 9 shows that for a fixed choice of $u_{*t}$
and $c_0$, even if they are "optimally" chosen, a portion of the measurements cannot be
represented by the model. Then, how does the saltation model perform if a single fixed $u_{*t}$ and
a single fixed $c_0$ are used as is often the case in aeolian models? The $p(Q)$ computed using the
model and derived from the JADE measurements are shown for $Q_{1min}$ and $Q_{30min}$ in Fig. 11. The
model is applied to estimate the saltation flux for individual particle size groups using the
optimally estimated $u_{*t}$ and $c_0$ (with $r_{u*t} = 1.12$ and $r_{c0} = 1.04$ for $Q_{1min}$, and $r_{u*t} = 1.04$ and $r_{c0}$
$= 0.92$ for $Q_{30min}$) and the total saltation flux is computed by integration over all particle size
groups, i.e., using Equation (3). Fig. 11 shows that for this option, the model over predicts the
probability of large $Q$, but under predicts the probability of small $Q$, in both cases of $Q_{1min}$ and
$Q_{30min}$. Obviously, to better reproduce the $Q_{1min}$ and $Q_{30min}$ pdfs, more values of $r_{u*t}$ and $r_{c0}$
sampled from the parameter pdfs are required. We have therefore modelled $Q_{1min}$ with other
choices of $r_{u*t}$ (1.12 and 0.56) and $r_{c0}$ (2.08, 0.01) and plotted the corresponding $Q_{1min}$ pdfs as
well as the averaged $Q_{1min}$ pdf of the three simulations. Similarly, we performed $Q_{30min}$ model
simulations with other $r_{u*t}$ (1.04) and $r_{c0}$ (1.84) values and examined the $Q_{30min}$ pdfs. With the
additional choices of the $r_{u*t}$ and $r_{c0}$ values, the $Q_{1min}$ and $Q_{30min}$ pdfs can be better reproduced.

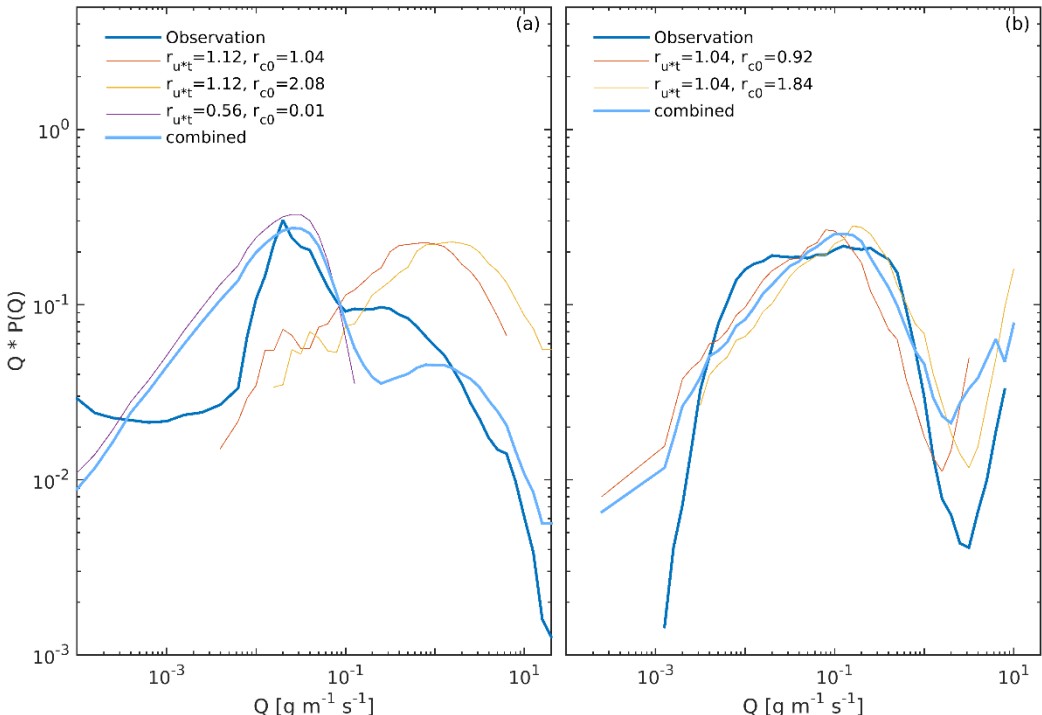

Figure 11: (a) Probability density functions of observed $Q$ and simulated $Q$ for 1-min averages
with several choices of $r_{u*t}$ and $r_{c0}$; (b) as (a), but for 30-min averages.

## 5. Summary


In this paper, we used the JADE data of saltation fluxes (resolution one second) and frictional
velocity (resolution one minute) to analyze the statistical behavior of turbulent saltation and
estimate the probability distribution of two important parameters in a saltation model, namely,
the threshold friction velocity, $u_{*t}$, and saltation coefficient, $c_0$.

Saltation fluxes show rich variations on different scales. It is found that while the widely used
$Q \sim u_*^3$ relationship holds in general, it can vary significantly between different wind erosion
events. In several wind erosion events observed in JADE, saltation hysteresis occurred. We
examined the probability density function of the saltation fluxes, $p(Q)$, and found that it
generally behaves like $Q^{-\alpha}$ with $\alpha \sim 1$. For $Q_{1sec}$, there is a distinct change in $\alpha$ at $Q = 3 \sim 4$ g m$^{-1}$ s$^{-1}$ with $\alpha \sim 1$ for smaller $Q$ and $\alpha \sim 4.0$ larger $Q$. It is shown that $p(Q)$ is dependent on the
averaging time intervals as a consequence of saltation intermittency.

We introduced the saltation intermittency functions $\gamma_a(u_{*t})$, $\gamma_b(u_*)$ and redefined saltation
intermittency $\gamma_c$ as the fraction of time during which saltation occurs at a given point in a given
time period, and computed these saltation intermittency measures using the JADE saltation flux
measurements. It is found that $\gamma_a(u_{*t})$ is one at $u_{*t} = 0$ and decreases to zero at about $u_{*t} = 0.5$
ms$^{-1}$. For $u_{*t} = 0.2$ ms$^{-1}$, $\gamma_a$ is 0.35. For $Q_{1min}$, , $\gamma_b(u_*)$ increases from about 0.6 at $u_* \sim 0.1$ ms$^{-1}$
to about one at $u_* = 0.3$ ms$^{-1}$. This shows that a considerable fraction of the saltation fluxes
occurs at small friction velocity and saltation is more intermittent under weak wind conditions
and is almost non-intermittent for $u_* > 0.3$ m s$^{-1}$. It is found that $\gamma_b(u_*)$ increased with decreasing
$u_*$ for $u_* < 0.1$ ms$^{-1}$ which is unexpected. Overall, $\gamma_c$ is found to be around 0.73. We computed
$\gamma_c$ as function of particle size and found that $\gamma_c(d)$ decreases with $d$, i.e., the saltation of larger
particles is more intermittent. Also, $\gamma_c(d)$ increases with increased averaging time intervals,
implying that the small scales features of turbulence play an important role in intermittent
saltation.
The power spectra of $Q$ and $u_*$ are found to have qualitatively similar behaviour. Both have a
maximum at about 10$^{-5}$ Hz, a minimum at about 10$^{-4}$ Hz and another peak at about 2x10$^{-3}$ Hz.
The maximum at 10$^{-5}$ Hz is related to the diurnal to synoptic events that drive wind erosion
episodes, the minimum at 10$^{-4}$ Hz is due to the lack of turbulent wind fluctuations at the time
scale of several hours, while the peak at 2x10$^{-3}$ Hz is caused by minute-scale gusts/large eddies
in turbulent flows. The power of the saltation rapidly decreases with frequency and becomes
relatively weak at frequencies of 0.1 Hz.
The posterior pdfs of the two parameters were estimated using the DREAM algorithm applied
to the JADE saltation flux measurements. While both $u_{*t}$ and $c_0$ have clear physical
interpretations, they are both stochastic parameters satisfying certain parameter pdfs. They also
dependent on the intervals of time averaging. Both $u_{*t}$ and $c_0$ for $Q_{1min}$ are larger than for $Q_{30min}$.
The pdf of $u_{*t}$ shows that it has a most frequent value close to the theoretical value, but can vary
over a range of 20% to 30%.The pdf of $c_0$ shows scatter over a wide range and it is unlikely
that a universal $c_0$ exists. In a saltation model, even if the optimally estimated $c_0$ is used,
considerable scatter between the model and the data would remain. The likely reason for the
stochasticity in $u_{*t}$ may be the temporal and spatial variations of particle cohesion, surface
roughness, particle shape etc. which cannot be well represented by a fixed deterministic value,
and the relatively large uncertainty in $c_0$ may be that this parameter depends on additional
factors (e.g. $u_*$ and soil particle size distribution) and is related to the fluctuations and
intermittency of saltation. It may also be that saltation in reality is never in equilibrium as
Bagnold (1941), Kawamura (1964) and Owen (1964) conceptualized, because due to turbulence,
sand grains are continuously entrained at different rates into the airflow and a continuous flow-
and particle-motion feedback takes place. As a consequence, it is difficult to treat $c_0$ as a
universal constant.
In this study, we highlighted the need to better understand saltation as a turbulent process and
the stochasticity of saltation model parameters. The concept of threshold friction velocity as a
stochastic variable was put forward in Shao (2001). Raffaele et al. (2016) examined the pdf of
$u_{*t}$ using data compiled from publications. Raffaele et al. (2018) studied how $u_{*t}$ uncertainties
propagate in saltation flux calculations and reported that in the case of small excess shear stress,
all models they tested amplify the uncertainty in estimated saltation flux, especially for coarse
sand. This finding is consistent with our notion that $c_0$ also is a stochastic variable. Due to the
stochasticity of the model parameters, the saltation model cannot reproduce the observation
even with the optimally estimated parameters (e.g. under estimation of weak saltation fluxes
and over estimation of strong saltation fluxes). A combination of several pairs of model
parameters appears to be required to reasonably reproduce the pdfs of saltation fluxes.
Our estimates of the parameter uncertainties is based on the data of a relatively simple aeolian
surface. For more complex surfaces, we expect the parameter uncertainties to be even more
pronounced.

**Acknowledgement:** This research is funded by the National Natural Science Foundation of China (No. 41571090, 41201539). The data used in this study were obtained in JADE (the Japan Australian Dust Experiment) by M. Ishizuka, M. Mikami, J. F. Leys, Y. Yamada, and S. Heidenreich. We are grateful to P. Schlüter and Q. Xia for support with data processing. We also wish to thank Dr. J. Gillies, Dr. M. Klose and an anomalous referee for their very helpful comments which prompt us to rework on a number of issues presented in the first version of the paper.

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
