# Peer review of "Dongwei Liu1, Masahid"

_Atmospheric Chemistry and Physics, 2017_

## Short Comment (SC1) · 11 Jan 2018

I found this to be an interesting paper that brings new insight into the relationship between the turbulent driving force of the wind and its links with the saltation system. I have included an edited version of the MS with suggested edits to improve the clarity and comments to address that will, when addressed, also aid in the clarity of the discussion.

---

## Referee Comment (RC1) · J. Gillies (Referee) · 12 Jan 2018

I found this to be an interesting paper that brings new insight into the relationship between the turbulent driving force of the wind and its links with the saltation system. I have included an edited version of the MS with suggested edits to improve the clarity and comments to address that will, when addressed, also aid in the clarity of the discussion.

Please also note the supplement to this comment:
https://www.atmos-chem-phys-discuss.net/acp-2017-1090/acp-2017-1090-RC1-supplement.pdf

[Figure]

**Supplement:**

[Figure]

[Figure]

**Turbulent Characteristics of Saltation and Uncertainty of Saltation Model Parameters**

Dongwei Liu[1], Masahide Ishizuka[2], Yaping Shao[3*]

[1]School of Ecology and Environment, Inner Mongolia University, China
liudw@imu.edu.cn
[2]Faculty of Engineering, Kagawa University, Japan
ishizuka@eng.kagawa-u.ac.jp
[3]Institute for Geophysics and Meteorology, University of Cologne, Germany
yshao@uni-koeln.de

**Abstract:** It is widely recognized that saltation is a turbulent process, similar to other transport
processes in the atmospheric boundary layer. But due to the lack of high frequency observations,
the statistic behavior of saltation is so far not well understood. In this study, we use the data
from the Japan-Australian Dust Experiment (JADE) to investigate turbulent saltation by
analyzing the probability density function, energy spectrum and intermittency of saltation
fluxes. Threshold friction velocity, $u_{*t}$, and saltation coefficient, $c_0$, are two important
parameters in saltation models, often assumed to be deterministic. But as saltation is turbulent,
we argue that it is more reasonable to consider them as parameters obeying certain probability
distributions. The JADE saltation fluxes are used to estimate the $u_{*t}$ and $c_0$ probability
distributions. The stochasticity of these parameters is attributed to the randomness in friction
velocity and threshold friction velocity as well as soil particle size.

**Keywords:** wind erosion; turbulent saltation; saltation intermittency; saltation model; threshold
friction velocity; saltation coefficient; maximum likelihood

**Highlight:** We use the data from a field experiment to investigate saltation by analysing the
probability density function, energy spectrum and intermittency of saltation fluxes. We also
estimate two key wind-erosion model parameters and their probabilistic distributions. It
continues the line of considering saltation as a turbulent process and represents a progress
towards deriving more general wind erosion models.

**1. Introduction**

It is known from the start of modern aeolian research [Bagnold, 1941] that saltation, the hop
motion of sand grains near the earth's surface, is a turbulent process. However, early aeolian
studies focused mainly on its "mean" behaviour. Most well-known is, for example the Owen
[Owen, 1964] saltation model which predicts that the vertically integrated saltation flux is
proportional to friction velocity cubed. A dedicated investigation on turbulent saltation was
conducted by Butterfield [1991]. Stout and Zobeck [1997] introduced the idea of saltation
intermittency and pointed out that even when the averaged friction velocity, $u_*$, is below the
threshold, saltation can still intermittently occur. The emphasis of the latter authors has been on
the saltation intermittency caused by the fluctuations of turbulent wind. Turbulent saltation has
attracted much attention in more recent years [e.g. McKenna Neuman et al. 2000; Davidson-
Arnott and Bauer, 2009; Sherman et al. 2017] and sophisticated models have been under
developed to model the process [e.g. Dupond et al. 2013]. However, due to the lack of high-
frequency field observations of saltation fluxes, the statistical behaviour of turbulent saltation
is to date not well understood.

[Figure]

A related problem is how saltation can parameterized in wind erosion models. For example, for
dust modelling, it is important to quantify saltation, as saltation bombardment is a main
mechanism for dust emission. In wind erosion models, threshold friction velocity, $u_{*t}$, is a key
parameter which depends on many factors including soil texture, moisture, salt concentration,
crust and surface roughness. In models, it is often expressed as

$$u_{*t}(d; \lambda, \theta, s_l, c_r, \ldots) = u_{*t}(d) f_\lambda(\lambda) f_w(\theta) f_{sc}(s_l) f_{cr}(c_r) \ldots \qquad (1)$$

where $u_{*t}(d)$ is the minimal threshold friction velocity for grain size $d$ [Shao and Lu, 2000]; $\lambda$ is
roughness frontal-area index; $\theta$ is   moisture; $s_l$ is soil salt content and $c_r$ is a descriptor of
surface crustiness; $f_\lambda, f_w, f_{sc}$ and $f_{cr}$   the corresponding correction functions. The corrections
are determined semi-empirically, e.g., $f_\lambda$ using the Raupach et al. [1993] scheme and $f_w$ the
Fécan et al. [1999] scheme. The corrections $f_{sc}$ and $f_{cr}$ are so far not well known.

[revised manuscript text omitted]

In JADE, three Sand Particle Counters (SPCs) [Yamada et al. 2002] were used to measure
saltation at the 0.05, 0.1 and 0.3 m levels with a sampling rate of 1 Hz. A SPC measures the
saltation of particles in the range of 38.9 - 654.3 µm in 32 bins with mean diameters of 38.9,
54.1, 69.2 etc. At each measurement height, the saltation flux density ($ML^{-2}T^{-1}$), $q$, is
obtained as the sum of $q_i$ (saltation flux for size bin $i$) for the 32 size bins, i.e.

$$q = \sum_{j=1}^{32} q_j \tag{11}$$

The saltation flux, $Q$, is then estimated by integrating $q$ over height, namely,

$$Q = \int q dz \tag{12}$$

In computing $Q$, we assume $q = q_0 \exp(-az)$ with $q_0$ and $a$ being fitted from the measurements.
As $q$ was measured only at three heights, the vertical resolution of $q$ is relatively poor and
inaccuracies in the $Q$ estimates are unavoidable. However, the profiles of $q$ are well behaved
and thus the inaccuracies in the $Q$ estimates are not expected to be so large to affect the
conclusions of this study.

Atmospheric variables, including wind speed, air temperature and humidity at various levels,
as well as radiation and precipitation, were measured using an automatic weather station (AWS).
Two anemometers were mounted at heights 0.53m and 2.16m on a mast for measuring wind
speed. Also available are the Monin-Obukhov length and sensible heat fluxes. From the wind
measurements, surface roughness length $z_0$ and friction velocity $u_*$ are derived, assuming a
logarithmic profile (with stability correction) of the mean wind. The roughness length for the
experiment site is estimated to be 0.48 mm. Observations of surface soil properties, including
soil temperature, soil moisture and surface cover were also made. The wind erosion model, as
detailed in Shao et al. (2011), is used for computing the saltation fluxes using the JADE
atmospheric and surface soil measurements as input. The essence of the saltation model
component is as described in Section 1. The fraction of erodible surface area, $\sigma_f$, used in
Equation (1), is estimated from photos using the technique as detailed Shao et al. (2011). For
the site, the fraction of surface cover is about 0.02, almost negligible.

The resolution of $Q$ is one second. We denote its time series as $Q_{1s}$. From $Q_{1s}$, the one-minute
averages, $Q_{1m}$ and 30-minute averages of saltation fluxes, $Q_{30m}$, are derived. The resolution of

friction velocity is one minute. We denote the one-minute averages of friction velocity as $u_{*1m}$
and the 30-minute averages $u_{*30m}$.

**3. Results**

**3.1 Statistical Features of Saltation**

Fig. 1 shows the time series of $Q_{1m}$ and $u_{*1m}$, and Fig. 2 $Q_{30m}$ and $u_{*30m}$. The figures show that
both $Q$ and $u_*$ significantly fluctuate, but the amplitude of $Q_{1m}$ fluctuations is several times of
that of $Q_{30m}$ fluctuations.

[Figure]

Figure 1: Observed time series of 1-min averaged saltation flux, $Q$ $(gm^{-1}s^{-1})$, and friction
velocity, $u_*$ $(ms^{-1})$. Note that the axes in (b) have different scales as in (a).

[Figure]

[Figure]

Figure 2: As Fig. 1, but for running means over 30-min intervals.

In Fig. 3, $Q$ is plotted against $u_*^3$. Several interesting features can be identified. For the majority
of the points, the $Q \sim u_*^3$ relationship appears to hold, but this relationship can vary significantly
even for the same data set from event to event. For example, large differences exist between
day 62 (a day of intensive wind erosion) and day 72 (a day of weak wind erosion), as seen in
both Fig. 3a and Fig. 3b. Also hysteresis can be observed in the  and
relationship (Fig. 3c): during an erosion event, for the same  saltation
is much stronger in the strengthening than in the weakening phase. There may be many reasons
for the hysteresis in the relationship between sediment flux and friction velocity but the most
likely are  differences in atmospheric turbulence (e.g. more gusty in the strengthening than
in the weakening phase) and time-varying surface conditions (e.g. particle sorting and
aerodynamic roughness).

[Figure]

Figure 3: (a) Saltation flux, $Q$ (gm$^{-1}$s$^{-1}$), plotted against friction velocity, $u_*^3$ (m$^3$s$^{-3}$), for 1-
minute averages; (b) As (a), but for 30-minute averages; (c) As (b), but enlarged to illustrated
saltation hysteresis.

How well the saltation model performs, whether $u_{*t}$ and $c_o$ are universal and how they are
probabilistically distributed must depend on the turbulent properties of saltation. As the JADE
saltation fluxes are sampled at 1 Hz, we can use the data to reveal (to some degree) the statistical
behavior of saltation. In Fig. 4, the pdfs of the saltation fluxes for different particle size groups
are plotted, computed using $Q_{1s}$ and $Q_{1m}$. It is seen that the pdfs generally behaves

$$p(Q) \propto Q^{-\alpha} \qquad\qquad\qquad (13)$$

[Figure]

In case of $Q_{1s}$, there seems to be a distinct change in $\alpha$ at a critical value of $Q_c \sim 3$ gm$^{-1}$s$^{-1}$, with
$\alpha = 0.8 \sim 0.9$ for $Q < Q_c$ and $\alpha = 4.0$ for $Q > Q_c$. The pdfs derived from $Q_{1m}$ appear to be
somewhat different, although the basic functional form is as given by Equation (13). In this
case, $\alpha$ is about 1 and drops off to about 2  $Q$ values. Fig. 4 shows that the pdfs of Q
depends quite significantly on the interval of time averaging. Fig. 4 also shows that after
averaging, smaller saltation fluxes become more likely. This is because the time series of $Q_{1s}$ is
more intermittent (see also Fig. 6).

[Figure]

Figure 4: (a) Probability density functions of saltation flux averaged over 1 second; (b) as (a),
but for saltation fluxes averaged over 1 minute.
In theory, $p(Q)$ can be derived from the pdf of $u_*$, $p(u_*)$. From Equation (2), we have

$$\frac{dQ}{du_*} = c_0 \frac{\rho}{g}\left(3u_*^2 + 2u_*u_{*_t} - u_{*_t}^2\right) \quad \text{for} \quad u_* > u_{*_t} \qquad (14)$$

It follows that

$$p(Q) = \begin{cases} p(u_*)\dfrac{du_*}{dQ} & \text{for} \quad u_* > u_{*_t} \\ 0 & \text{for} \quad u_* \le u_{*_t} \end{cases} \qquad (15)$$

Fig. 5a shows the $p(u_*)$ estimated from $u_{*1m}$ and Fig. 5b $p(Q)$ estimated from $Q_{1m}$. It is seen that
$p(u_*)$ can be well fitted with a Weibull distribution. We computed $p(Q)$ using Equation (15)
with the fitted $p(u_*)$, assuming $u_{*t} = 0.2$ ms$^{-1}$ and $c_o = 2.6$. It is seen that while the observed and
modelled $p(Q)$ only have qualitative similarities but are profoundly different. Fig. 5 shows that
even if the saltation model cannot reproduce the $p(Q)$ if $u_{*t}$. For example, the model fails to
predict the frequent weak saltation occurring when $u_*$ is below the specified threshold.

[Figure]

[Figure]

[Figure]

Figure 5: (a) Probability density functions of friction velocity, $p(u_*)$, plotted against $u_*$ (bars).
To compute $p(u_*)$, $u_{*1m}$ is used; a Weibull distribution (blue line) is fitted to $p(u_*)$; the red line
marks the assumed threshold friction velocity. (b) Probability density function of $Q$, $p(Q)$,
estimated using $Q_{1m}$ (blue) and using Equation (15) (black).

Also, the soil particle size distribution can influence $p(Q)$. In JADE, soil samples from the
experiment site were collected and the psds were analyzed in laboratory. Depending on the
methods used, the soil texture can be classified as sandy loam (clay 0.33%, silt 25% and sand
74.67%) or loamy sand (clay 11%, silt 35% and sand 54%). The soil at the observation site is
bimodal with one psd maximum at about 180 μm and another at about 500 μm (not shown).
The relatively large $p(Q)$ at about $Q_{1m} = 10^{-1}$ gm$^{-1}$s$^{-1}$ is related to the psd maximum at $d = 180$
μm.

Following Stout and Zobeck [1997], the intermittency of saltation, $\gamma_{int}$, is defined as the fraction
of time during which saltation occurs at a given point in a given time period. The latter authors
assumed that saltation is expected to occur only in the time windows when friction velocity
exceeds the threshold friction velocity. Therefore, suppose $p(u_*)$ is known, then $\gamma_{int}$ is

$$\gamma_{int} = 1 - \int_0^{u_{*t}} p(u_*)du_*$$

This definition of $\gamma_{int}$ is problematic, because $u_{*t}$ here is fixed. Stout and Zobeck [1997] used
the counts per second of sand impacts on a piezoelectric crystal saltation sensor as a measure
of saltation activity and found that $\gamma_{int}$ rarely exceeds 0.5.

We examined $\gamma_{int}$ using the JADE data. First, $\gamma_{int}$ is computed using $Q_{1m}$ conditionally sampled
for $u_* > u_{*c}$ with $u_{*c}$ successively varied from small to large. In Fig. 6a, $\gamma_{int}$ is plotted as a
function of $u_{*c}$. It is seen that on one-minute intervals, $\gamma_{int}$ has a maximum of about 0.25 for
small $u_{*c}$ and decreases to zero at about $u_{*c} = 0.3$ ms$^{-1}$. This shows that saltation intermittency
mainly occurs under weak wind conditions. If $\gamma_{int}$ is computed using $Q_{1s}$, then its maximum
reaches about 0.4, similar to that reported in Stout and Zobeck [1997]. For the one-second case,
we cannot plot $\gamma_{int}$ as a function of $u_{*c}$, because $u_*$ is not available at such high frequency. Fig.
6b shows (the maximum of) $\gamma_{int}$ as function of particle size for the one-second, one-minute and

30-minute cases. In general, $\gamma_{int}$ increases with particle size, i.e., the saltation of larger particles
is more intermittent. Also, $\gamma_{int}$ decreases with increased averaging time intervals, implying that
the small scales features of turbulence play an important role in intermittent saltation.

[Figure]

Figure 6: (a) Saltation intermittency, $\gamma_{int}$, computed using $Q_{1m}$ conditionally sampled for $u_* >$
$u_{*c}$; (b) $\gamma_{int}$ as a function of particle size for the one-second, one-minute and 30-minute cases.

Fig. 7 shows the power spectra of $Q$ and $u_*$ (Fig. 7a) as well their co-spectrum (Fig. 7b). The
power spectrum of $Q$ is computed using both $Q_{1s}$ and $Q_{1m}$, that of $u_*$ with $u_{*1m}$. It is seen that
the power spectra of $Q$ and $u_*$ have qualitatively very similar behaviour. Both have a maximum
at about $10^{-5}$ Hz, a minimum at about $10^{-4}$ Hz and another maximum at about $2 \times 10^{-3}$ Hz.  The
maximum at $10^{-5}$ Hz is related to the diurnal to synoptic events which drive the wind erosion
episodes, the minimum at $10^{-4}$ Hz is due to the lack of turbulent winds at the time scale of
several hours, while the maximum at $2 \times 10^{-3}$ Hz is caused by the minute-scale gusty winds/large
eddies in turbulent flows. Also the $Q$-$u_*$ co-spectrum shows that $Q$ and $u_*$ are most strongly
correlated on diurnal/synoptic and gust/large-eddy time scales. The saltation spectrum
computed using $Q_{1s}$ reveals again the maximum at $2 \times 10^{-3}$ Hz. However, the power of $Q$
spectrum rapidly decreases with frequency and become relatively weak on time scales smaller
than ~10 s.

[Figure]

Figure 7: (a) Normalized power spectrum of  (blue) computed with $u_{*1m}$,
together with the normalized power spectrum of saltation flux computed with $Q_{1m}$ (red) and $Q_{1s}$
(green). (b) Normalized $Q$-$u_*$ co-spectrum, computed using with $Q_{1m}$ and $u_{*1m}$. In both (a) and
(b), dots are unsmoothed spectra, while curves are smoothed spectra.

**4.2 Estimates of Saltation Model Parameters**

Given the turbulent nature of saltation, it is rational to treat $u_{*t}$ and $c_0$ in the saltation model
parameters obeying certain probability distributions. To examine the behavior of these
parameters, we introduce two coefficients $r_{c0}$ and $r_{u*t}$, and multiply them respectively to $c_0$ and
$u_{*t}$ in Equation (2). They are then varied to generate a model estimate of $Q$ using Equations (2)
and (3) with observed $u_*$ and the theoretical values of $u_{*t}$ and $c_0$. We denote the time series of
the modelled saltation flux as $Q_{M,i}$, ($i=1,N$) and of the corresponding measurement $Q_{D,i}$. The
absolute error, $\delta Q_A$, and Nash coefficient, $I_{Nash}$, are used as measures for the goodness of the
agreement between the model and the measurement. They are defined as,

$$\delta Q_A = \frac{1}{N}\sum|a_i|$$

$$I_{Nash} = (1 - \sum a_i^2 / \sum b_i^2)$$

with

$$a_i = Q_{M,i} - Q_{D,i}$$

$$b_i = Q_{M,i} - \frac{1}{N}\sum Q_{M,i}$$

$$c_i = \begin{cases} a_i / Q_{M,i} & Q_{M,i} \neq 0 \\ 0 & \text{else} \end{cases}$$

The prior pdfs of $r_{c0}$ and $r_{u*t}$ are assumed to be uniform. In the numerical experiment, we
randomly generate $r_{c0}$ and $r_{u*t}$ and seek their values, such that $\delta Q_A \leq \varepsilon$ and $I_{Nash} > \eta$. These

experiments are repeated for $Q_{1m}$ and $Q_{30m}$. The plots of $\delta Q_A$ and $I_{Nash}$ as functions of $r_{c0}$ and $r_{u*t}$
show that for certain values of $r_{c0}$ and $r_{u*t}$, the above conditions are satisfied. Fig. 8 shows that
for $Q_{1m}$, the best simulation is achieved with $r_{c0} = 1.23$ and $r_{u*t} = 1.05$, while for the $Q_{30m}$, with
$r_{c0} = 0.94$ and $r_{u*t} = 0.91$. This shows that while the "optimal" estimates of $u_{*t}$ and $c_0$ are close
to the corresponding theoretic values they are dependent on the time averaging intervals, with
both $u_{*t}$ and $c_0$ being larger for shorter averaging intervals.

[Figure]

Figure 8: $\delta Q_A$ and $I_{Nash}$ are both functions of $r_{c0}$ and $r_{u*t}$. Along the dashed curves, the
condition $\delta Q_A = \min$ is satisfied and along the solid curves the condition $I_{Nash} = \max$ is
satisfied. The curves are estimated with both one-minute and 30-minute averaged saltation
fluxes.

The parameter pdfs $p(r_{u*t})$ and $p(r_{c0})$ estimated using the DREAM algorithm are shown Fig. 9.
All pdfs are fitted to a $\Gamma$-distribution. As seen in Fig. 9a and 9c, the most frequent $r_{u*t}$ values
are respectively 1.12 and 1.04 for $Q_{1m}$ and $Q_{30m}$, close to the estimates of 1.05 and 0.91 found
in Fig. 8. For $Q_{1m}$, $r_{u*t}$ scatters in the range of ~1.12 ±0.2 and for $Q_{30m}$ in the range of ~1.04 ±
0.3. This implies that sometimes saltation occurs when $u_*$ is below the theoretic $u_{*t}$ value and
sometimes saltation does not occur even when $u_*$ is above the theoretic $u_{*t}$, as already seen in
Fig. 6a. In the case of $p(r_{c0})$ (Fig. 9c and 9d), the most frequent values of $r_{c0}$ for $Q_{1m}$ and $Q_{30m}$
are respectively 1.04 and 0.92, close to the optimal estimates of 1.23 and 0.94 found in Fig. 8.
But $r_{c0}$ scatters over a wide range, for instance, for $Q_{30m}$ between 0.5 and 5, i.e., $c_0$ is a rather
stochastic parameter.

[Figure]

[Figure]

[Figure]

Figure 9: (a) Parameter pdf $p(r_{u*t})$ for 1-min averaged saltation fluxes; (b) as (a), but for $p(r_{c0})$;
(c) and (d), as (a) and (b), but for 30-min averaged saltation fluxes.
In nature, many factors influence sediment transport, but the stochasticity of the parameters is
determined primarily by the turbulent fluctuations of friction velocity ( surface
shear stress), the randomness of threshold friction velocity, and soil particle size distribution
(representing particle response to forcing). Studies have shown, for instance, that small changes
in soil moisture can have large influences on saltation [Ishizuka et al. 2008] and soil moisture
in the very top soil layer can vary significantly over relatively short time periods. Over the
period of 18 days  soil moisture varied, this study, the influence of soil
moisture on saltation is accounted for via Equation (1) using the soil moisture measurements in
the top 0.05 m layer (see also Fig. 4a in Shao et al. 2011). The uncertainty in the wind erosion
parameters arising from soil moisture is most likely reflected in the stochasticity of $u_{*t}$.
The stochasticity of $c_0$ is more likely related to turbulence and particle size. To  this, we
divided the time series of the saltation fluxes into two subsets, one with $Q_{D,i} \leq 3$ gm$^{-1}$s$^{-1}$
representing weak saltation and one with $Q_{D,i} > 3$ gm$^{-1}$s$^{-1}$ representing significant saltation. This
separation is arbitrary but sufficient for making the point that wind erosion parameters depend
on $u_*$ which is a measure of turbulence intensity. The parameter pdfs, $p(r_{u*t})$ and $p(r_{c0})$, for the
subset $Q_{D,i} \leq 3$ gm$^{-1}$s$^{-1}$ is shown in Fig. 10. For $Q_{1m}$ and $Q_{30m}$, the most frequent $r_{u*t}$ values are
now respectively 0.99 and 0.85, somewhat smaller than the estimated values for the full set (see
Fig. 9). In comparison, the most frequent $r_{c0}$ values are now respectively 0.30 and 0.29, much
smaller than for the case when the full set is considered (see Fig. 9). This suggests that $c_0$ has a
dependency on $u_*$ and is smaller for smaller $u_*$ when saltation is more intermittent, as also seen
in Fig. 6a.

[Figure]

[Figure]

[Figure]

Figure 10: As Fig. 9, but estimated using the time series of saltation fluxes which satisfy $Q_{D,i} \leq$
$3$ gm$^{-1}$s$^{-1}$.
We fitted the pdfs, $p(r_{u*t})$ and $p(r_{c0})$, for individual particle size bins. It is found that the most
frequent $r_{u*t}$ values do not differ substantially among the particle sizes, but $r_{c0}$ depends
systematically on particle size. For example, the most frequent $r_{c0}$ values for 100.7, 151.2, 203.3,
314.5 and 397.7 µm are respectively 0.48, 1.31, 1.65, 3.06 and 4.00. These values are obtained
by first estimating $p(r_{c0})$ for the individual particle size bins with the measured saltation flux
for the corresponding bins and then normalizing $p(r_{c0})$ with the mass fraction of the size bins
of the parent soil. A least squares curve fitting shows that the most frequent $r_{c0}$ value depends
almost linearly on particle size:
$$r_{c_0} = 0.012d - 0.62 \qquad\qquad\qquad (16)$$
for the particle size range (100 to 400 µm) we tested, with $d$ being particle size in µm.
We have shown that both $u_{*t}$ and $c_0$ satisfy certain pdfs which depend on the properties of the
surface, atmospheric turbulence and soil particle size. Fig. 9 shows that for a fixed choice of $u_{*t}$
and $c_0$, even if they are "optimally" chosen, a portion of the measurements cannot be
represented by the model. Then, how does the saltation model perform if a single fixed $u_{*t}$ and
a single fixed $c_0$ are used as is often the case in aeolian models? The $p(Q)$ computed using the
model and derived from the JADE measurements are shown for $Q_{1m}$ and $Q_{30m}$ in Fig. 11. In this
case, the saltation model is applied to the individual particle size groups and the total (particle-
size integrated) saltation flux is computed using the $u_{*t}$ and $c_0$ optimally estimated. Fig. 11
shows that the model over predicts and probability of large $Q$, but under predicts the probability
of small $Q$ in both cases of $Q_{1m}$ and $Q_{30m}$.

[Figure]

[Figure]

[Figure]

Figure 11: (a) Probability density functions of observed $Q$ and simulated $Q$ for 1-min averages;
(b) as (a), but for 30-min averages.

**5. Summary**

In this paper, we  used the JADE data of saltation fluxes (resolution one second) and
(resolution one minute) to analyze the statistical behavior of turbulent
saltation and estimate the probability distribution of two of the most important parameters,
namely,  $u_{*t}$, and saltation coefficient, $c_0$, in a saltation model.

Saltation fluxes show a rich variations on different scales. It is found that while the widely used
$Q \sim u_*^3$ relationship holds in general, it can vary significantly between different wind erosion
events. In several wind erosion events observed in JADE, saltation hysteresis occurred. We
examined the probability density function of the saltation fluxes, $p(Q)$, and found that it
generally behaves like $Q^{-\alpha}$. For $Q_{1s}$, there is a distinct change in $\alpha$ at $Q = 3 \sim 4$ $\mathrm{gm^{-1}s^{-1}}$ with $\alpha =$
$0.8 \sim 0.9$ for smaller $Q$ and $\alpha = 4.0$ larger $Q$. It is shown that $p(Q)$ is dependent on the averaging
time intervals as a consequence of saltation intermittency.

We defined saltation intermittency, $\gamma_{int}$, as the fraction of time during which saltation occurs at
a given point in a given time period, and computed $\gamma_{int}$ using the JADE saltation flux
measurements. For $Q_{1m}$ conditionally sampled with $u_* > u_{*c}$, it is found that $\gamma_{int}$ has a maximum
of about 0.25 for small $u_{*c}$ and decreases to zero at about $u_{*c} = 0.3$ $\mathrm{ms^{-1}}$. This shows that saltation
intermittency mainly occurs under weak wind conditions. The $\gamma_{int}$ computed using $Q_{1s}$ has a
maximum of about 0.4. We have also computed $\gamma_{int}$ as a function of different particle sizes and
found that $\gamma_{int}$ in general increases with particle size.

The power spectra of  and  are found to have qualitatively similar
behaviour. Both have a maximum at about $10^{-5}$ Hz, a minimum at about $10^{-4}$ Hz and another
at about $2\times10^{-3}$ Hz. The maximum at $10^{-5}$ Hz is related to the diurnal to synoptic
events  drive wind erosion episodes, the minimum at $10^{-4}$ Hz is due to the lack of turbulent
wind fluctuations at the time scale of several hours, while the  at $2\times10^{-3}$ Hz is caused

by  minute-scale large eddies in turbulent flows. The power of the saltation
rapidly decreases with frequency and become relatively weak at frequencies of 0.1 Hz.
The posterior pdfs of the two parameters  estimated using the DREAM algorithm applied to
the JADE saltation flux measurements. While both $u_{*t}$ and $c_0$ have clear physical interpretations,
they appear to be dependent on the intervals of time averaging. Both $u_{*t}$ and $c_0$ for the 1-min
averages are larger than for the 30-min averages. The pdf of $u_{*t}$ shows that it has a most frequent
value close to the theoretic value, but can vary  a range of 20 to 30%. Therefore, the use of
the most frequent value of $u_{*t}$ in the saltation model seems to be reasonable. In contrast, the pdf
of $c_0$ shows  scatters over a much wider range. This suggests that it is rather unlikely that
a universal $c_0$ exists and the use of the most frequent value of $c_0$ would not reduce the scatter
between the model and the data. The likely reason for the relatively large uncertainty in $c_0$ may
be that it is parameter depending on additional factors (e.g.  and soil particle
size distribution). It may also be that saltation in reality is never in equilibrium as Bagnold
(1941), Kawamura (1964) and Owen (1964) conceptualized, because due to turbulent
fluctuations, sand grains are continuously entrained at different rates into the airflow and a
continuous flow and particle-motion feedback takes place. As a consequence, it is difficult to
treat $c_0$ as a universal constant.

**Acknowledgement:** This research is funded by the National Natural Science Foundation of
China (Control mechanism of groundwater-soil-vegetation continuum on dust emission in
desert playas, No. 41571090). The data used in this study  obtained in JADE (the Japan
Australian Dust Experiment) by M. Ishizuka, M. Mikami, J. F. Leys, Y. Yamada, and S.
Heidenreich. We are grateful to P. Schlüter and Q. Xia for support with data processing.

**References:**

[revised manuscript text omitted]

---

## Author Comment (AC1) · 17 Jan 2018

Dear Dr Gilles,

many thanks for your careful reading of the manuscript and very details comments. We will consider your suggestions carefully and update the manuscript soon. Also a few other readers have provided suggestions by sending me emails. Their comments will also be reflected in the updated version.

Regards,

Yaping Shao (on behalf of authors)

[Figure]

2017.

---

## Referee Comment (RC2) · Anonymous Referee #2 · 1 Feb 2018

Review on the manuscript "Turbulent characteristics of saltation and uncertainty on saltation model parameters" By Dongwei Liu, Masahide Ishizuka and Yaping Shao

General comment : This paper aims at investigating the turbulent behaviour of the saltation flux based on experimental data from a previous field campaign (JADE Experiment, Ishizuka et al., 2008, 2014). From these measurements, the authors questioned the relevance of the models used to compute the saltation and dust fluxes in 3D models that described an average behaviour and ignore the small-scale dynamical features. The author propose to re-visit the determination of the coefficients used in such "average" saltation models to account for these effects. The subject is of great interest and the data set used by the authors offers a unique opportunity to investigate the impact of the turbulence on the total saltation fluxes and its variation with the particle size. However, the text is too succinct in many parts of the manuscript, some key information are missing and the conclusions drawn at each step of the work are not sufficiently stated and argued. The links between the different parts of the manuscript are not sufficiently explicit. I would thus suggest the authors to provide a deeper analysis of their results and to further improve the text and wording to make their work more convincing.

Major comments : Introduction : The introduction clearly state the position of the problem. Some suggestions to better organize the text are given in "Minor comments".

Part 2 : I would suggest to separate by subtitled the first part describing the computation method and the one describing the data and their pre-treatment. In fact the reading could be more easy if the data were described first and the computing method after.

A few lines to introduce the objective of the part concerning the computing algorithm and to make the link with the introduction are absolutely required. Several method are briefly described to end up with the one selected by the authors without arguing why this method is better adapted than the others to analyse saltation and wind friction velocities data sets. At the end of the chapter (page 4 line 141) the reader does not really know what is computed with this method regarding to the different results presented in the following parts.

On line 163-164, the author mention that the fitting of the vertical profile lead to inaccuracies in the estimation of Q, but that it would not affect the results of this study. A quantitative estimation of these accuracies is needed.

The authors used a data set of U* average over one minute. A discussion on the relevance of this time-scale would be welcome. U* is more commonly averaged over tens of minutes to represent the average effect of the main turbulent structures.

Part 3 : This part should be divided in subsections (time series and wind dependence of Q; Pdf; intermittency; power spectra). In general, the figures and interpretations

given in part 3 are not sufficiently described and commented to be fully understood and appreciate.

a) Time series : Figures 1 and 2 shows times series of Q and U*, with a 12-days data set and a zoom on a two days data sets that is not included in the previous 12-days period. These figures (b) are not commented in the text and at this stage of the manuscript, the reader cannot understand why they are shown. Page 13, line 203-213 : the behaviour of Q is very different on day 71 and 72 and the authors argued that the hysteresis behaviour during these two days can be due to changes in surface properties and atmospheric turbulence. Is there any observational evidences for these differences or is it just speculative? If the atmospheric turbulence is different, one may expect different results for these two specifics days in the following parts of the paper. But they are no more evoked in the following.

b) Probability density functions : Figure 4 present the probability density function of the saltation fluxes for different particles sizes. How does the pdf of the total flux compare to the pdf of the size-segregated fluxes ?

The results concerning the pdf of the wind friction velocity and of Q is very questionable. The "modelled" Q is computed after fitting a Weibull function on the experimentally determined U*. Why isn't it computed directly from the experimental wind friction velocity? The authors argued that the Weibull function fits "well" the U* pdf, but the quality of the fitting does not appear to be so good on figure 5 : the number of wind speeds just above the threshold seems to be significantly underestimated while the highest winds seems to be over estimated by this function. Why not fitting only the values above the threshold or fitting U*3? This may improve the representation of the pdf and the quality of the modelled Q. The poor level of agreement between the computed and measured Q is also surprising since the correlation between the modelled and measured Q was of 0.7 for the same experiment and the same model (Shao et al., 2011).

The discussion on the impact of the soil size distribution (page 8 lines 263-269) is

not clear neither the conclusion that can be drawn. Could the impact of the soil size distribution on the modelled flux be estimated since it is an input data of the Shao's model?

c) Intermittency : The "Intermittency section" should include a more precise description on the way it is computed. Indeed, the fact that it is as low as 0.1 when the threshold is 0.2 m/s does not seem consistent with figure 1 : for the well identified saltation events (days 56, 57, 60, 61, 62, 63, 69) the saltation flux Q1m looks positive when u* is higher than 0.2. A lower value suggest that the intermittency is computed over the whole time series, i.e. including periods of high winds with no saltation. Integrating periods of high winds with and without saltation does not corresponds to the initial concept on intermittency which correspond to the fact that during a given event, the wind velocity can be successively below or above the saltation threshold. From one avent to the other many factors can act to prevent wind erosion on a given day compared to the others (precipitations, soil moisture). A table providing, event by event, the number of time steps with u*>u*c and the fraction of these time steps with Q>0 would make things more clear. The way the lower limit for Q is defined should also be described.

Figure 6 shows that the intermittency vary with the particle size and the authors conclude that the saltation of larger particle is more intermittent. An explanation could be the saltation threshold increases with the particle size (at least for particle diameter >80-100 $\mu$m)

d) Power spectra : The power spectra of the saltation flux and of the wind friction velocity is one of the most interesting result of the manuscript. The way it is computed should be described and the results further discussed and analysed.

It is quite common in the literature on turbulence to see normalized power spectrum of the wind velocity, including both the horizontal and the vertical components measured by sonic anemometers. The frequency is also often normalized to the height of measurements and the mean wind speed, which allows to compare the results from

different sites. Here the authors show the power spectrum of the wind friction velocity as a function of the frequency of measurements. They should explain why and how they produce the results from figure 7. How should the power spectrum of the wind friction velocity compare to the "classical" power spectrum of wind velocity? The authors comment the behaviour of the spectrum for different frequencies and relate this to the typical time scale of dynamical processes. References to similar results in terms of wind spectrum would make the results more convincing. The figure also raises the question of the data set of Q used to compute the power spectrum. The scale of the frequencies extend down to 10-6, i.e. more than 270 days while the whole sampling period is less than one month. From figure 1, it seems that the saltation episodes do not last longer than a day. Are the data set for Q1m and Q1s limited to periods for which the measured Q(z) are non-null (and once again the way the minimum Q is defined should be described) or do they include periods with no saltation recorded ?

The similarity of the spectrum of Q and U* is a strinking results that should be further highlighted. The power spectra of Q1m and Q1s both exhibit a peak at 2.10-3 Hz (less than 10 min). What does this mean? That a 1min acquisition time step is sufficient to properly describe the way saltation is impacted by turbulence ? This is also an original results that should be further discussed.

Part 4.2 : The objective of this part is to test whether a probability distribution of u*t and c0 would improve the capability of the saltation model to reproduce the measured fluxes. This part also suffer for a lack of description on the method to estimate the pdf of ru*t and rc0 and on the way the modelled Q are finally computed for the final comparison with the measured Q. In this comparison, rather that the modelled and measured pdf of Q, one would expect a quantification of the benefit on the level of agreement between the measured and computed Q (correlation coefficients, RMSE, for example). It would be interesting also to test the change in the level of agreement with observations using the full distribution of the r parameters (figure 9) and the peak value only.

The author discuss the possible influence of the soil moisture, but the conclusion is not clear : the sentence "over the period .." does not seems to be correct (a verb missing ?) and cannot be understood. It is not clear from the following sentence ("in this study .;") whether the influence of the measured soil moisture is effectively accounted for in the modelled Q used to determine the distributions of r.

The discussion of the stochasticity of c0 in particular for weak saltation is not sufficiently linked to the discussion on the intermittency, which is mentioned only at the very end of the section.

From figure 6, it is expected that u*t may vary with the particle size. But only rc0 is found dependent on particle size. The authors should comment on this possible contradiction. They state that the most frequent values of ru*t do not differ substantially, but what about the parameters of the distribution ? And what range of variation is considered as substantial ? When the author described the way rc0 is determined, it is not clear how they combine the determination of ru*t with the determination of rc0. Once again a more precise description should be given before the presentation and discussion the results.

In the last section, it should be clearly specified how the computation is made : are the measured u*t used ? is the soil moisture effect included ? are the "optimally estimated" u*t and c0 corrected with the p(u*t) and p(rc0) ? If all these effects are included, what are the main sources of differences between the measured and modelled Q ? Does the level of agreement between the modelled and measured pds of Q depend on the erosion events ? What about the saltation flux cumulated for the different erosion events ? Depending on the application, the error could be acceptable, but in any case, it should be quantified.

Part 5 : Even if a few line of conclusion and perspective are given at the end of this section, I would suggest a to add more conclusive elements and some perspectives open by the presented work in terms of modelling but also in terms of improving the

experimental set up for the coming field experiment.

Minor comment :

Page 1 line 38 : Replace Staut by Stout.

Page 2 line 68-69 : It should be stated that, beside the establishment of the flux equations, the value of the coefficient is generally derived from measurements.

Page 2 line 69-70 : I would suggest to skip a line before the sentence "the total (all particle size) saltation flux . . .". Since the size dependence of the flux equation was not proposed by Kawamura (1964) nor White (1979) but was mainly added for modelling applications.

Page 2 line 75 : I would suggest to skip a line before the sentence "Observations show, .." and to add references from the literature to give a range of c0 derived from observations.

Page 4 line 167-172. The temporal resolution of the atmospheric variable measurements should be given here.

Page 4 line 174-177 : I am not sure it is the right place to present the wins erosion model.

Page 7 figure 4, please specify Q1s and Q1m on the axes of the figure and use the scale scale for Q to make the two figures easily comparable.

Page 8, line 285-286. "This shows that saltation intermittency mainly occurs under weak wind conditions" : since intermittency is defined as the fraction of time the wind friction velocity exceed the threshold, isn't it obvious that it occurs mainly when the wind friction velocity os close to the threshold ?

Page 10 line 318, part "4.2" should be part "3.2".

Page 13, line 406-408 : Figure 9 reports the distribution of rc0 and ru*t, it does not

show that "for a fixed choice of u*t and co, even if they are optimally chose, a portion of the measurements cannot be represented by the model".

---

## Author Response (AR1)

Dear Editors of Atmospheric Chemistry and Physics,

On behalf of the co-authors (Liu, Ishizuka, Mikami and Shao), I wish to thank the two referees for their very helpful comments and also two readers who send us their comments. These comments are now considered in the revised version of the paper for your consideration. The point by point reply and the revised manuscript are uploaded.

Please address all correspondence to:

Prof. Yaping Shao
Institute for Geophysics and Meteorology, University of Cologne, Germany
Tel: + 49 (0) 221 470-3688
Fax: + 49 (0) 221 470-5161
E-mail: yshao@uni-koeln.de (preferred contact address)

**Reply to SC2:**

We wish to thank SC2 for her efforts to work through our paper and providing very helpful comments. Our reply to her comments are as follows:

Comment to Fig 3: thanks for this very good suggestion. We will slightly change the graph to make the hysteresis clearer

Comment to Fig 4: As suggested, we will modify the figure

L247: Accepted

L268-269: Yes. This can be seen from size solved $Q$ data

L378-387: Thanks to SC2 for this comment, in which she stated that "*I think making this conclusion here is somewhat problematic, because it is based on data sampled for Q < 3 g/m/s. Although Q depends on u\* and therefore small Q are likely to coincide with small u\*, u\* might not be the only reason for Q to be small. Therefore, sampling for small Q might introduce a bias by selecting only those Q that already tend to have small c0, in particular around 3 g/m/s, the selected cut-off. The result of smaller c0 for smaller fluxes can therefore in my opinion not unambiguously be used to prove a dependence of c0 on u\*/turbulence.*"

This discussion motivated us to think deeper about the process of saltation and we would like to retain our argument in the text. Basically, weak saltation occurs in case of smaller friction velocity. We now know that for smaller friction velocity, saltation becomes gradually more intermittent. Therefore, c0, a description of the relation between time averaged saltation flux and time averaged friction velocity becomes smaller. We added a sentence and hope it becomes clearer.

L410-412: Clarified.

General comments: SC2 made two general comments as follows: "I have two general comments/questions:
(1) I wonder whether there might be a (small) temporal delay between measured winds and the associated measured Q_1s which could depend on particle size (due to the particles' inertia) and which might have an effect on the parameter results. Perhaps this could be worth exploring, even if only to rule it out. Due to the necessary temporal integration of u∗, this is likely invisible though (if present at all). (2) How do you think the parameter PDFs would change for a different (perhaps less ideal) surface? I think that a brief discussion on that would be very interesting."

Due to data limitation, we do not have shear stress data with one second resolution. Consequently, we were unable to check the correlation of shear stress and sand drift at frequency of 1 Hz. The question rated by SC2 is certainly important, which we will investigate with better experiment design and instrumentation. Our data show that the two quantities are well correlated at the frequencies of large eddies and synoptic events, a pronounced phase shift between the two quantities is so far not identified. Earlier studies (e.g. Butterfield, 1991) suggest that the response time of the aeolian surface is about 1 second, therefore, we do not think there are phase differences between saltation flux and shear stress on time scales over one minute or longer.

We thank SC2 for this comment. We will add a paragraph of our view on the problem. We will also cite two recent papers by Raffaele et al. (2016; 2018). The added paragraph will be as follows:

In this study, we highlighted the need to better understand parameter uncertainty in saltation models and the processes responsible for the uncertainty. The concept of threshold friction velocity as a stochastic variable was first proposed in Shao (2001). Raffaele et al. (2016) more systematically examined the probabilistic distribution of $u_{*t}$ using data compiled from earlier publications. Raffaele et al. (2018) then studied how $u_{*t}$ uncertainties propagate in saltation flux calculations and reported that in the case of small excess shear stress, all models they tested amplify the uncertainty in estimated saltation flux, especially for coarse sand. This finding is consistent with our notion that $c_0$ also is a stochastic variable. Our estimate of the parameter uncertainties is based on the data of a relatively simple aeolian surface. For more complex surfaces, we expect the parameter uncertainties to be even more pronounced.

**RC1:** We wish to thank Dr. Gilles for his efforts for working through our draft and providing very helpful comments and very detailed editorial suggestions. Most of his editorial suggestions are accepted, as reflected in the revised manuscript. Our reply to his other comments are as follows:

L44: We kept our original formulation to stress that variations in threshold can also lead to intermittent saltation.

We accepted all his editorial comments and answered all his queries. In particular, we did more work to Fig. 11.

**RC2:** We are grateful to RC2 for his/her constructive comments. We feel encouraged that RC2 finds our work "of great interest" and we find his/her critical comments accurate and very helpful. In the revised manuscript, we have tried to accommodate these comments. Our point by point reply is as follows:

Major comments:

Introduction: The introduction clearly state the position of the problem. Some suggestions to better organize the text are given in "Minor comments".

**We modified the text according to the minor suggestions.**

Part2: I would suggest to separate by subtitled the first part describing the computation method and the one describing the data and their pre-treatment. In fact the reading could be more easy if the data were described first and the computing method after.

A few lines to introduce the objective of the part concerning the computing algorithm and to make the link with the introduction are absolutely required. Several method are briefly described to end up with the one selected by the authors without arguing why this method is better adapted than the others to analyse saltation and wind friction velocities datasets.

At the end of the chapter (page4 line 141) the reader does not really know what is computed with this method regarding to the different results presented in the following parts.

**We thank the referee for the comments. We have substantially reworked on the section and hope the description is now clearer.**

Online163-164, the author mention that the fitting of the vertical profile lead to inaccuracies in the estimation of Q, but that it would not affect the results of this study. A quantitative estimation of these accuracies is needed.

**The request of the referee is understood, but unfortunately, we are not able to give a more detailed statement on the absolute accuracy of the Q measurements using the SPCs. Care was taken such that individual SPC works properly (e.g. wind-tunnel calibration), but as measurements were only made at 3 levels, the profile of saltation flux density was under represented. However, as our study is mainly on temporal variations of saltation fluxes, the inaccuracy in the absolute values should not significantly alter our conclusions. We slightly modified the text.**

The authors used a data set of U* average over one minute. A discussion on the relevance of this time-scale would be welcome. U* is more commonly averaged over tens of minutes to represent the average effect of the main turbulent structures.

**This is a challenging question, as there is really no standard for how long one should average wind to "correctly" estimate $u_*$, but we can answer the question from three perspectives. First, if u* is used as a scaling velocity for turbulence properties in atmospheric boundary layer, e.g., turbulence intensity, eddy diffusivity, M-O similarity functions etc., it seems necessary to average over sufficiently long time to obtain a more or less "constant" shear stress and $u_*$. Second, if u* is merely a surrogate for shear stress and one is interested in the variations of the shear stress, then shorter averaging times are justified, subject to the condition that the response of aeolian fluxes to shear stress is faster. We know (roughly) from earlier studies (Butterfield, 1991; Anderson and Haff, 1988) that the response time of aeolian fluxes**

in turbulent flows is of the order of one second. Third, to derive meaningful shear stress from wind profile, what averaging wind data do we have to use? This depends on whether the assumptions of flow steady state and horizontal homogeneity are satisfied. The JADE site is a flat farm land, such that the use of wind profile data for deriving shear stress for 1-minute intervals can be justified. We added a paragraph to this effect in the revised manuscript.

Part3: This part should be divided in subsections (time series and wind dependence of Q; Pdf; intermittency; power spectra). In general, the figures and interpretations given in part 3 are not sufficiently described and commented to be fully understood and appreciate.

We went through the text and tried to add clarifications.

a) Time series: Figures 1 and 2 shows times series of Q and U*, with a 12-days data set and a zoom on a two days data sets that is not included in the previous 12 Interactive days period. These figures (b) are not commented in the text and at this stage of the manuscript, the reader cannot understand why they are shown.

We reformulated the section and made changes to the graphs. The purpose of this section is to show the time series this study is based on and discuss some turbulent features which we can identify directly by looking at the times series.

Page 13, line 203-213: the behavior of Q is very different on day 71 and72 and the authors argued that the hysteresis behavior during these two days can be due to changes in surface properties and atmospheric turbulence. Is there any observational evidences for these differences or is it just speculative? If the atmospheric turbulence is different, one may expect different results for these two specifics days in the following parts of the paper. But they are no more evoked in the following.

There is some evidence for this. We substantially changed the text and added Fig. 3d to Fig. 3 showing the time series of ($u_{*1min} - u_{*30min}$) as a measure of turbulent fluctuations. It is seen that saltation is usually not only associated with high surface shear stress but also with high shear stress fluctuations. The profound difference in the $Q \sim u_*$ relationship between D70-71 and D72 (Fig. 3b) can be attributed to the strong differences in turbulent fluctuations between them: D70-71 was a hot and gusty day with top (2 cm) soil temperature reaching 53$^o$C, while D72 was a cooler and less gusty day with soil temperature about 5$^o$C lower. It seems that saltation hysteresis cannot be simply attributed to turbulence. We speculate that it is more likely to be related to flow-saltation feedbacks (e.g. stronger splash entrainment in the strengthening phase) and the modification of surface aerodynamic conditions (e.g. particle sorting and reduced surface roughness Reynolds number). But we need an extra study to fully answer the question.

b) Probability density functions: Figure4 present the probability density function of the saltation fluxes for different particles sizes. How does the pdf of the total flux compare to the pdf of the size-segregated fluxes? The results concerning the pdf of the wind friction velocity and of Q is very questionable. The "modelled" Q is computed after fitting a Weibull function on the experimentally determined U*.Why isn't it computed directly from the experimental wind friction velocity? The authors argued that the Weibull function fits "well" the U*pdf, but the quality of the fitting does not appear to be so good on figure 5: the number of wind speeds just above the threshold seems to be significantly underestimated while the highest winds seems to be overestimated by this function. Why not fitting only the values above the threshold or fitting U*3? This may improve the representation of the pdf and the quality of the modelled Q. The poor level of agreement between the computed and measured Qis also surprising since the correlation between the modelled and measured Q was of 0.7 for the same experiment and the same model (Shao et al., 2011).

The pdf of the total flux is later shown in Fig. 5, but we now added this also to Fig. 4.

With Fig. 5, we try to understand the behavior of the pdfs of the saltation fluxes. Basically, qualitatively, the rapid drop of the probability of strong saltation is caused by that of u*. But, quantitatively, the model cannot reproduce the pdf. While this observation seems obvious, when we plotted the results, but still, the information is useful. We now followed the suggestions of the referee and have plotted additional fittings to the pdf of u*.

The poor level of agreement between the computed and measured Q is also surprising since the correlation between the modelled and measured Q was of 0.7 for the same experiment and the same model (Shao et al., 2011).

In Shao et al. (2011) only one event was studied, as by that time, the saltation fluxes was not completely computed for all events.

The discussion on the impact of the soil size distribution (page 8 lines 263-269) is not clear neither the conclusion that can be drawn. Could the impact of the soil size distribution on the modelled flux be estimated since it is an input data of the Shao's model?

These sentences are removed from the text to avoid confusion.

c) Intermittency: The "Intermittency section" should include a more precise description Interactive on the way it is computed. Indeed, the fact that it is as low as 0.1 when the threshold is comment 0.2m/s does not seem consistent with figure 1: for the well identified saltation events (days 56, 57, 60, 61, 62, 63, 69) the saltation flux Q1m looks positive when u* is higher than 0.2. A lower value suggest that the intermittency is computed over the whole time series, i.e. including periods of high winds with no saltation. Integrating periods of high winds with and without saltation does not corresponds to the initial concept on intermittency which correspond to the fact that during a given event, the wind velocity can be successively below or above the saltation threshold. From one event to the other many factors can act to prevent wind erosion on a given day compared to the others (precipitations, soil moisture). A table providing, event by event, the number of time steps with u*>u* cand the fraction of these time steps with Q>0 would make things more clear. The way the lower limit for Q is defined should also be described. Figure6 shows that the intermittency vary with the particle size and the authors conclude that the saltation of larger particle is more intermittent. An explanation could be the saltation threshold increases with the particle size (at least for particle diameter >80-100 m)

We understand the points the referee tried to make. We substantially reworked on this section, by introducing new definitions of intermittency. Our preference is to have some understanding of the statistic behavior of intermittency. We have therefore not focused on the intermittency of individual events, as we would end up with a lot of different values which would be difficult to interpret.

d) Power spectra: The power spectra of the saltation flux and of the wind friction velocity is one of the most interesting result of the manuscript. The way it is computed should be described and the results further discussed and analysed. It is quite common in the literature on turbulence to see normalized power spectrum of the wind velocity, including both the horizontal and the vertical components measured by nic anemometers. The frequency is also often normalized to the height of measurements and the mean wind speed, which allows to compare the results from different sites. Here the authors show the power spectrum of the wind friction velocity as a function of the frequency of measurements. They should explain why and how ACPD they produce the results from figure 7. How should the power spectrum of the wind friction velocity compare to the "classical" power spectrum of wind velocity?

The authors comment the behavior of the spectrum for different frequencies and relate this Interactive to the typical time scale of dynamical processes. References to similar results in terms comment of wind spectrum would make the results more convincing. The figure also raises the question of the data set of Q used to compute the power spectrum. The scale of the frequencies extend down to 10-6, i.e. more than 270 days while the whole sampling period is less than one month. From figure 1, it seems that the saltation episodes do not last longer than a day. Are the data set for Q1m and Q1s limited to periods for which the measured Q(z) are non-null (and once again the way the minimum Q is defined should be described) or do they include periods with no saltation recorded?

We have improve this section. As our sampling rate is relatively low, it is difficult to directly compute our u* spectrum with the Reynolds stress spectrum. We have nevertheless added references in which Reynolds shear stress spectra are shown. The frequency of $10^{-6}$ Hz corresponds to a period of 11.6 days. The time series of the data is about twice that length. The data points for which all sensors gave Q = 0 are included in the power spectra computation. The data points, for which no measurements were reported by the sensor were excluded in the computation. As the Q fluxes cover irregular time intervals, a non-uniform Discrete Fourier Transform (NDFT) is used. This is not a standard Matlab function, but we have tested that in the limit of regular time series our program delivers the some results. We are therefore confident that the power spectra analysis is correct.

The similarity of the spectrum of Q and U*is a strinking results that should be further highlighted. The power spectra of Q1m and Q1s both exhibit a peak at 2.10-3 Hz (less than 10 min). What does this mean? That a 1min acquisition time step is sufficient to properly describe the way saltation is impacted by turbulence? This is also an original results that should be further discussed.

The consistency between power spectra of $Q_{1min}$ and $Q_{1sec}$ at low frequency is expected, as $Q_{1min}$ is derived from $Q_{1sec}$. We are not sure about the suggestion that "a 1min acquisition time step is sufficient to properly describe the way saltation is impacted by turbulence". What it shows is that one-minute sampling is sufficient to resolve the impact of very large eddies, but not turbulence.

Part 4.2 : The objective of this part is to test whether a probability distribution of u*t and c0 would improve the capability of the saltation model to reproduce the measured fluxes. This part also suffer for a lack of description on the method to estimate the pdf of ru*t and rc0 and on the way the modelled Q are finally computed for the final comparison with the measured Q. In this comparison, rather that the modelled and measured pdf of Q, one would expect a quantification of the benefit on the level of agreement between the measured and computed Q (correlation coefficients, RMSE, for example). It would be interesting also to test the change in the level of agreement with observations using the full distribution of the r parameters (figure 9) and the peak value only.

We have tried to clarify how rc0 and ru*t are computed. Yes. We have tried to use a full set of rc0 and rustar values to illustrate the improvement.

The author discuss the possible influence of the soil moisture, but the conclusion is not clear: the sentence "over the period .." does not seems to be correct (a verb missing ACPD ?) and cannot be understood. It is not clear from the following sentence ("in this study .;") whether the influence of the measured soil moisture is effectively accounted for in the modelled Q used to determine the distributions of r.

This is now clarified.

The discussion of the stochasticity of c0 in particular for weak saltation is not sufficiently linked to the discussion on the intermittency, which is mentioned only at the very end of the section.

We have tried to clarify

From figure 6, it is expected that u*t may vary with the particle size. But only rc0 is found dependent on particle size. The authors should comment on this possible contradiction. They state that the most frequent values of ru*t do not differ substantially, but what about the parameters of the distribution? And what range of variation is considered as substantial? When the author described thewayrc0is determined, it is not clear how they combine the determination of ru*t with the determination of rc0. Once again a more precise description should be given before the presentation and discussion the results.

We had some inaccurate descriptions re. ru*t close to the theoretic value. We have now modified the text. The dependency of rc0 on particle size is due to the different intermittency of different particle size groups. We have added some lines in the test about this,

In the last section, it should be clearly specified how the computation is made: are the measured u*t used? is the soil moisture effect included? are the "optimally estimated" u*t and c0 corrected with the p(u*t) and p(rc0)? If all these effects are included, what are the main sources of differences between the measured and modelled Q? Does the level of agreement between the modelled and measured pdfs of Q depend on the erosion events? What about the saltation flux cumulated for the different erosion events? Depending on the application, the error could be acceptable, but in any case, it should be quantified.

The effect of soil moisture has been considered. It is now clear that as the parameters u*t and c0 are distributed, a model using a fixed u*t and c0 cannot reproduce the measurements, but only "optimally" reproduce the measurements as defined by The absolute error, $\delta Q_A$, and Nash coefficient, $I_{Nash}$, which are used as measures for the goodness of the agreement between the model and the measurement. But major aspects of the measurements cannot be reproduced with deterministic u*t and c0, for example, the pdf of the Q fluxes.

Part5: Even if a few line of conclusion and perspective are given at the end of this section, I would suggest a to add more conclusive elements and some perspectives open by the presented work in terms of modelling but also in terms of improving the experimental setup for the coming field experiment.

Thanks for this comment. It makes good sense. We modified the conclusion section.

Minor comment:

Page1 line-38: Replace Staut by Stout.

**We corrected the mistake.**

Page2 line 68-69: It should be stated that, beside the establishment of the flux equations, the value of the coefficient is generally derived from measurements.

Accepted.

Page2 line 69-70: I would suggest to skip a line before the sentence "the total (all particle size) saltation flux :::". Since the size dependence of the flux equation was not proposed by Kawamura (1964) nor White (1979) but was mainly added for modelling applications.

Page 2 line-75: I would suggest to skip a line before the sentence "Observations show, .." and to add references from the literature to give a range of c0 derived from observations.

We did not do the separations, as we would otherwise ended up with several very short paragraphs. As to c0, we cannot give a range, as relatively few studies use the Kawamura scheme. We however added Gillette (1997) and Leys (1998) as references. Their data imply that c0 can vary over orders (may be two) in magnitude, as the data of Leys (1998) here show.

[Figure]

Figure R1: Fitting the Owen saltation model with observed data. The measurements were made on four soils with different textures denoted A, B, C and D, which corresponds to the U.S. taxonomy Aridosol (agrid), Aridosol (calic orthidf), Vertisol and Aridisol (haplargid). Two treatments were applied to each soil: bare uncultivated (denoted n) and bare cultivated (denoted c), giving a total of 8 soil-treatment combinations. The parameter r2 gives an indication for the goodness of the fitting with a perfect fit having a value of 1 (from Leys, 1998).

Page4 line 167-172. The temporal resolution of the atmospheric variable measurements should be given here.

Accepted

Page4 line 174-177: I am not sure it is the right place to present the wind erosion model.

This model has been discussed in detail in Shao (2011) and elsewhere. The relevant module consists basically Equations (1) – (3) already given I Section 1. We therefore think the information given is sufficient. In the revised draft, we say more explicitly that the model used consists of Equations (1) – (3), and we added the sentences how particle size distribution is done.

Page7 figure 4, please specify Q1s and Q1m on the axes of the figure and use the scale for Q to make the two figures easily comparable.

Accepted.

Page 8, line 285-286. "This shows that saltation intermittency mainly occurs under weak wind conditions": since intermittency is defined as the fraction of time the wind friction velocity exceed the threshold, isn't it obvious that it occurs mainly when the wind friction velocity is close to the threshold?

We think it is nice that the data confirm what one would expect, but we are not sure without seeing the data that this is obvious, as intermittency must depends on the pdf of $u_*$ near the surface. i.e., how turbulent flow is. We did not change anything in the text in this regard.

Page 10 line 318, part "4.2" should be part "3.2".

Thanks. Changed

Page 13, line 406-408: Figure 9 reports the distribution of $r_{c0}$ and $r_{u_{*t}}$, it does not show that "for a fixed choice of $u_{*t}$ and co, even if they are optimally chose, a portion of the measurements cannot be represented by the model".

Thanks for this comment. A lot of changes have been made to this part of the text. We believe the concern of the referee is now adequately addressed.

Sincerely,

Yaping Shao
(on behalf of co-authors)

[revised manuscript text omitted]

$$p(Q) \propto Q^{-\alpha} \qquad\qquad (11\cancel{3})$$

In case of $Q_{1sec}$, there seems to be a distinct change in $\alpha$ at a critical value of $Q_c \sim 3$ g m$^{-1}$-s$^{-1}$,
with $\alpha \sim 1 = 0.8 \sim 0.9$ for $Q < Q_c$ and $\alpha \sim = 4.0$ for $Q > Q_c$. The pdfs derived from $Q_{1min}$ appear
to follow be somewhat different, although the basic functional form of is as given by Equation
(11\cancel{3}). Again, In this case, $\alpha$ is about 1 and tends to be larger drops off to about 2 for large $Q$
values. Fig. 4 shows that the pdfs of $Q$ depends quite significantly on the interval of time
averaging, i.e., . Fig. 4 also shows that after averaging, smaller saltation fluxes become more
frequent likely. This is because the time series of $Q_{1sec}$ is more intermittent (see also Fig. 6).

[Figure]

[Figure]

Figure 4:  Probability density functions of $Q_{1sec}$ (solid lines) and of $Q_{1min}$ (dashed lines) for four different particle sizes. Two additional lines $p(Q) \sim Q^{-1}$ and $Q^{-4}$ are drawn as reference.

The pdfs of $Q_{1sec}$ and $Q_{1min}$ integrated over all particles are shown in Figure 5b. Again, the pdfs show the general behavior of $p(Q) \sim Q^{-1}$. In theory, $p(Q)$ can be derived from the pdf of $u_*$, $p(u_*)$. From Equation (2), we have

$$\frac{dQ}{du_*} = c_0 \frac{\rho}{g} \left( 3u_*^2 + 2u_* u_{*_t} - u_{*_t}^2 \right) \quad \text{for} \quad u_* > u_{*_t} \qquad (1\underline{2}4)$$

This can be used to obtain

$$p(Q) = \begin{cases} p(u_*)\dfrac{du_*}{dQ} & \text{for} \quad u_* > u_{*_t} \\ 0 & \text{for} \quad u_* \leq u_{*_t} \end{cases} \qquad (13\text{5})$$

Fig. 5a shows the $p(u_*)$ estimated from $u_{*1min}$ together with the fitted Weibull distribution. For
the fitting, emphasis is made to ensure that $p(u_*)$ for $u_* > 0.2$ ms$^{-1}$ is best approximated. and Fig.
5b shows the $p(Q)$ estimated from $Q_{1min}$. It is seen that $p(u_*)$ can be well fitted with a Weibull
distribution. We computed $p(Q)$ using Equation (13\text{5}) with the fitted $p(u_*)$, assuming $u_{*t} = 0.2$
ms$^{-1}$ and $c_0 = 2.6$. It is seen that while the observed and modelled $p(Q)$ only have qualitative
similarities (namely $p(Q)$ decreases with increasing $Q$) but using Equations (12) and (13) we
are profoundly different. Fig. 5 shows that even if the saltation model cannot well reproduce
the observed $p(Q)$ if $u_{*t}$. For example, the model fails to predict the lowly frequent strong
saltation fluxes and fails to predict the highly frequent –weak saltation occurring when $u_*$ is
below the specified threshold. Tests using several smaller $u_{*t}$ values (0, 0.05 and 0.1). With
smaller $u_{*t}$ values, the highly frequent weak saltation fluxes are better reproduced, but fra from
satisfactory.

[Figure]

[Figure]

Figure 5: (a) Probability density functions of friction velocity, $p(u_*)$, plotted against $u_*$ (bars). To compute $p(u_*)$, $u_{*1min}$ is used; a Weibull distribution (blue line) is fitted to $p(u_*)$; the red line marks the assumed threshold friction velocity. (b) Probability density function of $Q$, $p(Q)$, estimated using $Q_{1min}$ (blue) and $Q_{1sec}$ (dark red) and using Equation (13) assuming several u*t values ($u_{*t}$ = 0.0 m s$^{-1}$, green; 0.05 m s$^{-1}$, red; 0.1 m s$^{-1}$, yellow; 0.2 m s$^{-1}$, black). The $p(Q) \sim Q_{-1}$ line is also drawn for comparison.

Also, the soil particle size distribution can influence $p(Q)$. In JADE, soil samples from the experiment site were collected and the psds were analyzed in laboratory. Depending on the methods used, the soil texture can be classified as sandy loam (clay 0.33%, silt 25% and sand 74.67%) or loamy sand (clay 11%, silt 35% and sand 54%). The soil at the observation site is bimodal with one psd maximum at about 180 μm and another at about 500 μm (not shown). The relatively large $p(Q)$ at about $Q_{1m}$ = 10$^{-1}$ gm$^{-1}$s$^{-1}$ is related to the psd maximum at $d$ = 180 μm.

**3.3 Saltation Intermittency**

Following Stout and Zobeck [1997], the intermittency of saltation, $\gamma_{int}$, is defined as the fraction of time during which saltation occurs at a given point in a given time period. It should be pointed out that as saltation is a turbulent process, saltation intermittency describes only the behaviour of the process at $u_* \sim u_{*t}$, i.e., saltation intermittency is merely a special, although important, case of turbulent saltation. Several formulations of $\gamma$ are possible. Stout and Zobeck [1997] The latter authors assumed that saltation is expected to occurs only in the time windows when $u_*$ friction velocity exceeds the $u_{*t}$ threshold friction velocity. Therefore, if suppose $p(u_*)$ is known, then $\gamma$ $\gamma_{int}$ for a given $u_{*t}$ can be estimated as is

$$\gamma_a(u_{*t}) = 1 - \int_0^{u_{*t}} p(u_*)du_* \qquad \gamma_a = 1 - \int_0^{u_{*tc}} p(u_*)du_* \qquad (14a)$$

This definition of $\gamma_{int}$ is problematic, because $u_*$ here is fixed. Stout and Zobeck [1997] used the counts per second of sand impacts on a piezoelectric crystal saltation sensor as a measure of saltation activity and found that $\gamma_{a int}$ rarely exceededs 0.5.

In Equation (14a)  $u_{*t}$  is fixed and thus
saltation intermittency is attributed entirely to the fluctuations of $\mu_*$. In reality, $\mu_{*t}$ also fluctuates
and satisfies certain pdfs (Raffaele et al., 2016). In analogy to Equation (14a), $\gamma$ for a given $\mu_*$
can be estimated as

$$\gamma_b(u_*) = 1 - \int_{u_*}^{\infty} p(u_{*t})du_{*t} \qquad\qquad (14b)$$

More generally, we can define saltation intermittency as

$$\gamma_c = \int_0^{\infty}\left[1 - \int_0^{u_{*t}} p(u_*)du_*\right]p(u_{*t})du_{*t} = \int_0^{\infty}\gamma_a(u_{*t})p(u_{*t})du_{*t} \qquad (14c)$$

or

$$\gamma_c = \int_0^{\infty}\left[1 - \int_{u_*}^{\infty} p(u_{*t})du_{*t}\right]p(u_*)du_* = \int_0^{\infty}\gamma_b(u_*)p(u_*)du_* \qquad (14d)$$

Equations (14c) and (14d) reduce to Equation (14a) in case of $p(u_{*t}) = \delta(u_{*t})$ and to Equation
(14b) in case of $p(u_*) = \delta(u_*)$, respectively.

The computation of saltation intermittency function $\gamma_a(u_{*t})$ is done by integrating $p(u_*)$ (Fig. 5a)
to fixed value of $\mu_{*t}$. In Fig. 6a, $\gamma_a$ as function of $\mu_{*t}$ is plotted. The behaviour of $\gamma_a(u_*)$ is as
expected: it is one at $\mu_{*t} = 0$ and decreases to zero at about $\mu_{*t} = 0.5$ ms$^{-1}$ as in the case of JADE,
$\mu_*$ rarely exceeded this value. For $\mu_{*t} = 0.2$ ms$^{-1}$, $\gamma_a$ is 0.35, comparable with the result of Stout
and Zobeck (1997) who reported an intermittency of 0.4. As $p(u_{*t})$ is not known, Equation (14b)
cannot be used directly, but we can  compute  $\gamma_b(u_*)$  using the JADE data. First,
it  is computed using $Q_{1min}$  This is done for $u_* > u_{*c}$
by selecting a fixed $\mu_*$ say $\mu_{*c}$, and counting the time
fraction, $T_{u*}$, which satisfies $|u_* - u_{*c}| < \delta$, (used is $\delta = 0.05$ ms$^{-1}$) and  time fraction, $T_{Q1min}$,
which satisfies $|u_* - u_{*c}| < \delta$ and $Q_{1min} > 0$. By definition, saltation intermittency is $T_{Q1min}/T_{u*}$,
The value $\mu_{*c}$ is successively increased to obtain saltation intermittency function $\gamma_b(u_*)$,
as plotted in  Fig. 6a.
It is seen that for $Q_{1min}$  $\gamma_{bint}(u_*)$, increases from about 0.6 at $\mu_*$
$\sim 0.1$ ms$^{-1}$ to about one at $\mu_* = 0.3$ ms$^{-1}$
This shows that in JADE a considerable fraction of the saltation
fluxes was recorded at u* below the perceived threshold friction velocity (about 0.2 ms$^{-1}$),
saltation is more intermit under weak wind conditions and becomes non-
intermittent for $u_* > 0.3$ ms$^{-1}$. The  of $\gamma_b(u_*)$ with decreasing $\mu_*$ for $\mu_* < 0.1$ ms$^{-1}$ is
however unexpected. The expected $\gamma_b(u_*)$ for small $\mu_*$ is as depicted using the dashed line. A
likely reason for the unexpected behaviour of $\gamma_b(u_*)$ is that during a wind erosion event, grains
in saltation may continue to hop even when $\mu_*$ is temporarily reduced to small values. The
uncertainty in the data also needs to be considered, as the sample size for determining the ratio
$T_{Q1min}/T_{u*}$ becomes smaller. More complete datasets are required to answer these questions.
Finally, $\gamma_c$ is computed by using Equation (14d) and is found to be around
0.73. For the one-second case, we cannot plot $\gamma_{bint}$ as a function of $u_{*c}$, because u* is not available
at such high frequency. We computed $\gamma_c$ for individual particle size groups (Fig. 6b) using $Q_{1sec}$,
$Q_{1min}$ and $Q_{30min}$, which is the time fraction of saltation for a given particle size, $d$, during the
saltation event.

It is found that  $\gamma_a(d)_{int}$ increases with $d$,
i.e., the saltation of larger particles is more intermittent. Also, $\gamma_b(d)_{int}$ decreases with
increased averaging time intervals, implying that the small scales features of turbulence play an
important role in intermittent saltation.

[Figure]

Figure 6: (a) Saltation intermittency function  $\gamma_a(u_{*t})_{int}$, and $\gamma_b(u_*)$. See text for more
details. (b) $\gamma_{c\ int}$ as a function of particle
size for $Q_{1sec}$, $Q_{1min}$ and $Q_{30min}$.
**3.4 Spectrum of Saltation Fluxes**
Spectral analysis is a widely used for characterising the variations of a stochastic process on
different scales. Using the JADE data, we computed the power spectrum of saltation fluxes,
$P_Q(f)$ at frequency $f$, and of friction velocity, $P_{u*}(f)$, using a non-uniform discrete Fourier transform. For comparison, the power spectra are normalized with the respective variances of
the signal. In atmospheric boundary-layer studies, the spectra of various turbulence quantities
have been thoroughly investigated (Stull, 1988). Examples for spectra of Reynolds shear stress
can be found in McNaughton and Laubach (2000). Fig. 7 shows $P_Q(f)$ and the power spectra of
$Q$ and $P_{u*}(f)$ (Fig. 7a) as well their co-spectrum (Fig. 7b). $P_Q(f)$ The power spectrum of $Q$ is
computed using both $Q_{1sec}$ and $Q_{1min}$, and $P_{u*}(f)$ that of $u_*$ with $u_{*1min}$. It is seen that the power
spectra of $Q$ and $u_*$ have qualitatively very similar behaviour. Both have a maximum at about
$10^{-5}$ Hz, a minimum at about $10^{-4}$ Hz and another peak maximum at about $2x10^{-3}$ Hz.  The
maximum at $10^{-5}$ Hz is related to the diurnal patterns and changing to synoptic events, which
drive the wind erosion episodes, the minimum at $10^{-4}$ Hz is due to the lack of turbulent winds
at the time scale of several hours, while the peak maximum at $2x10^{-3}$ Hz is caused by the minute-
scale gusty winds/large eddies in turbulent flows. Also the $Q$-$u_*$ co-spectrum shows that $Q$ and
$u_*$ are most strongly correlated on diurnal/synoptic and gust/large-eddy time scales. $P_Q(f)$ The
saltation spectrum computed using $Q_{1sec}$ reveals again the peaks at $10^{-5}$ Hz and maximum at
$2x10^{-3}$ Hz. The However, the power of the $Q$ spectrum then rapidly decreases with frequency.

[revised manuscript text omitted]